# RECONVIAGEN: TOWARDS ACCURATE MULTI-VIEW 3D OBJECT RECONSTRUCTION VIA GENERATION

**Jiahao Chang**[1,2*]    **Chongjie Ye**[2,1*]    **Yushuang Wu**[2]    **Yuantao Chen**[1]
**Yidan Zhang**[1]    **Zhongjin Luo**[2]    **Chenghong Li**[2]    **Yihao Zhi**[1]    **Xiaoguang Han**[1,2,3†]
[1]School of Science and Engineering, The Chinese University of Hong Kong, Shenzhen
[2]Shenzhen Future Network of Intelligence Institute
[3]Guangdong Provincial Key Laboratory of Future Networks of Intelligence

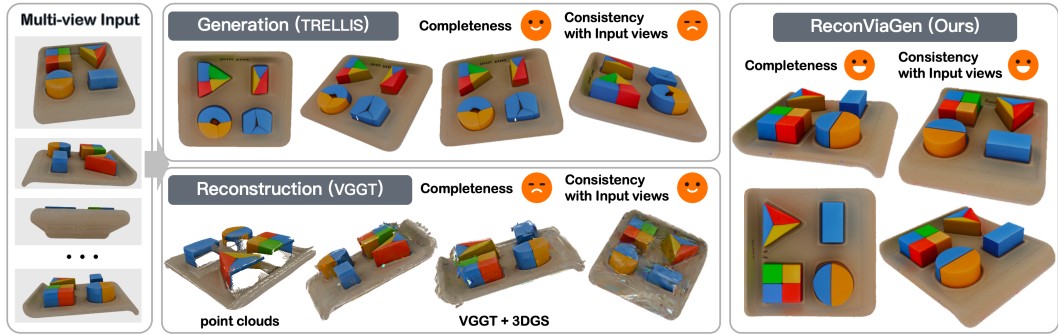

Figure 1: In the task of 3D object reconstruction from multi-view images, existing pure reconstruction methods can only produce incomplete results, while generation-based methods can get plausible complete results but with strong inconsistency with input images. Our ReconViaGen integrates 3D reconstruction and diffusion-based generation priors into one framework that leads to accurate reconstructions.

## ABSTRACT

Existing multi-view 3D object reconstruction methods heavily rely on sufficient overlap between input views, where occlusions and sparse coverage in practice frequently yield severe reconstruction incompleteness. Recent advancements in diffusion-based 3D generative techniques offer the potential to address these limitations by leveraging learned generative priors to "hallucinate" invisible parts of objects, thereby generating plausible 3D structures. However, the stochastic nature of the inference process limits the accuracy and reliability of generation results, preventing existing reconstruction frameworks from integrating such 3D generative priors. In this work, we comprehensively analyze the reasons why diffusion-based 3D generative methods fail to achieve high consistency, including (a) the insufficiency in constructing and leveraging cross-view connections when extracting multi-view image features as conditions, and (b) the poor controllability of iterative denoising during local detail generation, which easily leads to plausible but inconsistent fine geometric and texture details with inputs. Accordingly, we propose ReconViaGen to innovatively integrate reconstruction priors into the generative framework and devise several strategies that effectively address these issues. Extensive experiments demonstrate that our ReconViaGen can reconstruct complete and accurate 3D models consistent with input views in both global structure and local details. Project page: https://jiahao620.github.io/reconviagen.

---

*Equal contribution.
†Corresponding Author.

# 1 INTRODUCTION

In the field of 3D computer vision, multiview 3D object reconstruction has long been a fundamental yet challenging task, with numerous applications in areas such as VR, AR, and 3D modeling. Existing multiview reconstruction methods typically rely on enough visual cues and learned correspondences between views to estimate the 3D structure and appearance of the object Mildenhall et al. (2020); Kerbl et al. (2023); Leroy et al. (2024); Wang et al. (2024a; 2025a). However, these methods often face significant limitations when dealing with weak-texture objects or incomplete image captures due to occlusions or the presence of support surfaces. As a result, reconstructed 3D models tend to have holes, artifacts, and missing/blurred geometric details, which severely restrict the reconstruction completeness He et al. (2024); Xu et al. (2024c).

Recent advances in diffusion-based 3D generative techniques have shown great promise in addressing these limitations. These techniques leverage 3D generative priors learned from large-scale 3D data to generate complete 3D outputs from sparse- or even single-view images Li et al. (2024b); Zhao et al. (2025); Li et al. (2025a); Zhang et al. (2024b); Ye et al. (2025). Such strong generative priors can effectively "hallucinate" the invisible portions of objects with plausible high-quality geometry and appearance, thereby showing great potential in 3D reconstruction by filling in the missing details and improving the completeness. However, the stochastic nature of the diffusion-based inference process introduces significant uncertainty and variability in the generated results, making it challenging to achieve high accuracy and reliability, especially the pixel-level alignment required in accurate reconstruction. This stochasticity has largely hindered the effective integration of diffusion-based 3D generative priors into existing multi-view reconstruction frameworks.

Pioneering explorations have been made in the field of 3D diffusion-based generation from multi-view images Xiang et al. (2024); Zhao et al. (2025). However, their predictions still suffer from inaccurate global structures and inconsistent local details. The inherent key reasons of the failure include (i) the insufficiency in constructing cross-view correlations when extracting multi-view image features as conditions, resulting in inaccurate estimation in both object geometry and texture, at the global and local level, (ii) the poor controllability and stability of the denoising process during inference, which easily results in inconsistency with input views especially in detailed geometry and texture estimation. To address these issues, we present ReconViaGen that innovatively integrates multi-view stereo priors into the diffusion-based generative framework for object reconstruction. Our solution includes three stages: (i) a pre-trained strong reconstructor Wang et al. (2025a) is developed to build a multi-view stereo understanding of the object geometry and texture, aggregated into a single global token list and a set of local token lists, for representing the global geometry and the detailed per-view appearance, respectively; (ii) a coarse-to-fine 3D generator Xiang et al. (2024) first estimates the coarse structure and then produces the fine textured mesh, under the conditioning of global and local tokens from the first stage, respectively; (iii) refining the estimated poses from the reconstructor using the generation from the second stage, and encouraging the pixel-wise alignment with input views using a novel rendering-aware velocity compensation mechanism, where input images coupled with estimated camera poses are used to explicitly guide the denoising trajectory of local latent representations.

Extensive experiments on the Dora-bench Chen et al. (2024) and OmniObject3D Wu et al. (2023b) datasets validate that our ReconViaGen can achieve state-of-the-art (SOTA) reconstruction performance in both global shape accuracy and completeness and local details in geometry and textures. Our contributions are summarized as follows:

- We propose a novel framework called ReconViaGen, which is the first to integrate strong reconstruction priors into a diffusion-based 3D generator for accurate and complete multi-view object reconstruction. A key design is to aggregate image features rich in reconstruction priors as multi-view-aware diffusion conditions.

- The generation adopts a coarse-to-fine paradigm, which leverages global and local reconstruction-based conditions to generate accurate coarse and then fine results in both geometry and texture. Additionally, a novel rendering-aware velocity compensation mechanism is proposed that constrains the denoising trajectory of local latent representations for detailed pixel-level alignment.

- Extensive experiments on the Dora-bench and OminiObject3D datasets are conducted that validate the effectiveness and superiority of the proposed ReconViaGen, which achieves SOTA performance.

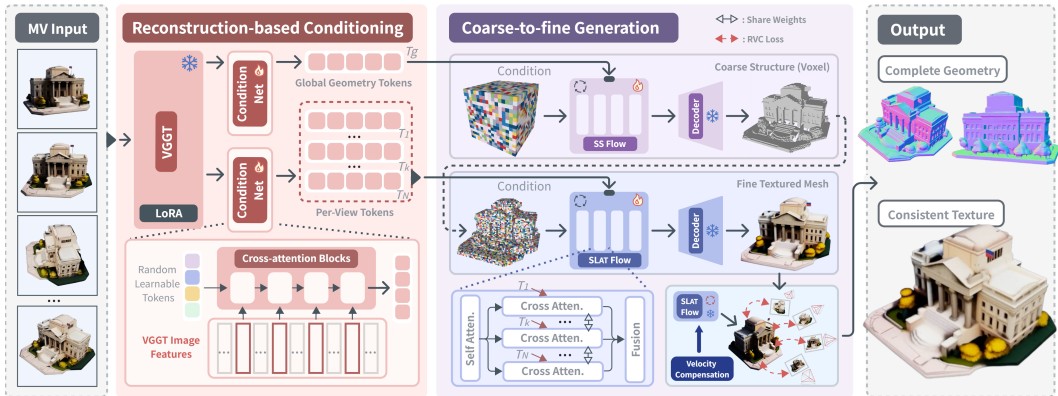

Figure 2: An overview illustration of the proposed ReconViaGen framework, which integrates strong reconstruction priors with 3D diffusion-based generation priors for accurate reconstruction at both the global and local level.

## 2 RELATED WORK

**Single-view 3D Generation.** Great developments have been made in single-view 3D Object Generation. Recent methods can be divided into two groups: 2D prior-based and 3D native generative methods. DreamFusion Poole et al. (2022) and its following successors Tang et al. (2024); Qiu et al. (2024); Wang et al. (2024b); Lin et al. (2023); Tang et al. (2023a) distill the 3D knowledge from pretrained 2D models. Another line of work develops multi-view diffusion based on pre-trained image or video generators and conducts view fusion for 3D outputs Li et al. (2023); Xu et al. (2024b); Tang et al. (2025); Xu et al. (2024d); Wang et al. (2025b); Wei et al. (2024); Liu et al. (2023a; 2024); Xu et al. (2024e); Li et al. (2024a); Zuo et al. (2024); Wu et al. (2024a). Differently, 3D native generative methods employ diffusion based on different 3D representations like point clouds Luo & Hu (2021); Zhou et al. (2021); Nichol et al. (2022), voxel grids Hui et al. (2022); Tang et al. (2023b); Müller et al. (2023), Triplanes Chen et al. (2023); Wang et al. (2023b); Shue et al. (2023), 3D Gaussians Zhang et al. (2024a). More recently, 3D latent diffusion has been explored to directly learn the mapping between the image and 3D geometry Zhang et al. (2023); Zhao et al. (2024); Li et al. (2024b); Zhang et al. (2024b); Wu et al. (2024b); Li et al. (2025a); Zhao et al. (2025); Ye et al. (2025), which greatly improves the generation quality. However, multi-view 3D generation is still under-explored, suffering from high variations in generation, easy inconsistency with input images, or strong reliance on input viewpoints Xiang et al. (2024); Zhao et al. (2025), which hinders their direct application in accurate 3D object reconstruction.

**Multi-view 3D Reconstruction.** Traditional methods conduct multi-view stereo (MVS) to reconstruct the visible surface of objects by triangulating correspondences across multiple calibrated images Furukawa et al. (2015); Galliani et al. (2015); Schönberger et al. (2016); Xu & Tao (2019). Learning-based MVS methods Yao et al. (2018; 2019); Chen et al. (2019); Cheng et al. (2020); Gu et al. (2020); Yang et al. (2020); Wang et al. (2021a) employ deep neural networks to enhance both reconstruction quality and computational efficiency. Scene-specific NeRF methods Lin et al. (2021); Wang et al. (2021b) adopt bundle adjustment from conventional SfM pipelines to jointly optimize camera parameters along with radiance field from dense views. Recently, DUSt3R Wang et al. (2024a) and its follow-up works Smart et al. (2024); Leroy et al. (2024); Wang et al. (2025a) together estimate point clouds and camera poses from paired or more views, which releases the reliance on camera parameters, but suffers from incomplete reconstruction results caused by the point cloud representation. Focusing on object reconstructions, large reconstruction models Hong et al. (2023) are explored to produce complete reconstructions via regressing a more compact or structured 3D representation (*e.g.* 3D Gaussians Kerbl et al. (2023) and Triplane) from multi-view inputs Wei et al. (2024); Xu et al. (2024b); Tang et al. (2025); Xu et al. (2024d), but requiring view inputs from certain camera poses. Follow-up methods further support pose-free reconstructions Wu et al. (2023a); Wang et al. (2023a); Jiang et al. (2023); He et al. (2024), while they tend to predict smooth and blurred details, especially in invisible regions. Differently, our method introduces diffusion-based 3D generation priors to advance pose-free object reconstruction in fidelity and completeness.

**Generative Priors in 3D Object Reconstruction.** Generative priors are introduced into 3D reconstruction frameworks to assist in predicting plausible geometries or textures in invisible portions of objects. Existing methods mainly introduce two kinds of priors: (i) diffusion-based 2D generative prior and (ii) regression-based 3D generative prior. The former is often used in single-view 3D reconstruction by generating plausible multi-view images first and conducting reconstruction Yang et al. (2024); Li et al. (2023; 2024a); Xu et al. (2024b); Tang et al. (2025); Xu et al. (2024d); Wang et al. (2025b); Wei et al. (2024); Liu et al. (2023a; 2024); Wu et al. (2024a). For pose-free sparse-view reconstruction, iFusion Wu et al. (2023a) leverages Zero123 Liu et al. (2023b) predictions within an optimization pipeline to align poses and generate novel views for reconstruction. However, the inconsistency between views still limits the performance of this pipeline. Regression-based 3D generative priors are introduced to regress a unified compact 3D representation, avoiding this issue, for example 3D neural volume Jiang et al. (2023), Triplane Hong et al. (2023); Wei et al. (2024); Wang et al. (2023a), and 3D Gaussians He et al. (2024); Xu et al. (2024c); Smart et al. (2024). Diffusion-based generative priors prove superior to regressive ones in generating detailed results in both geometry and texture Li et al. (2025a); Zhao et al. (2025); Zhang et al. (2024b); Ye et al. (2025). One2345++ and its follow-up work Liu et al. (2024); Xu et al. (2024a) develop 3D volume diffusion conditioned by multi-view inputs. However, 3D volume suffer from poor compactness, so a trade-off between the diffusion learning difficulty and representation capability limits their performance. Differently, our method builds upon strong diffusion-based 3D generative priors Xiang et al. (2024), with powerful reconstruction priors Wang et al. (2025a) constraining the denoising process for accurate 3D outputs of high-fidelity details.

## 3 METHODOLOGY

### 3.1 PRELIMINARY

Given a set of $N$ uncalibrated multi-view images of an object $I = \{I_i\}_{i=1}^N$, the task of pose-free multi-view reconstruction aims to obtain the complete 3D object $O$. Our framework leverages two kinds of strong priors to achieve complete and accurate reconstruction results: the reconstruction prior from VGGT Wang et al. (2025a) and the generation prior from TRELLIS Xiang et al. (2024). In this section, we first introduce these two priors as preliminaries.

**Reconstruction prior of VGGT** VGGT Wang et al. (2025a) achieves SOTA results in pose-free multi-view 3D reconstruction, providing a powerful reconstruction prior. It adopts a feed-forward transformer architecture designed for efficient and unified 3D scene reconstruction from single/multiple images. Multi-view images $I$ are first fed into a DINO-based ViT Oquab et al. (2024) simultaneously for tokenization and feature extraction into $\phi_{\text{dino}}$. Then, 24 self-attention layers further address $\phi_{\text{dino}}$ into 3D-aware features $\{\phi_i\}_{i=1}^{24}$ with an alternating attention strategy, switching between frame-wise and global self-attention to balance local and global information and enhance multi-view consistency. Finally, four prediction heads decode the output of 4 layers (4-th, 11-th, 17-th, and 23-rd), *i.e.*, $\phi_{\text{vggt}}(I) = \{\phi_4, \phi_{11}, \phi_{17}, \phi_{24}\}$, into camera parameters, depth map, point map, and tracking feature predictions. To adapt to object reconstruction, we fine-tune VGGT on an object-reconstruction dataset (see Sec. 4.1 for details). A LoRA fine-tuning on the VGGT aggregator is employed to preserve the pre-trained 3D geometric priors, with a multi-task objective:

$$\mathcal{L}_{\text{VGGT}}(\theta) = \mathcal{L}_{\text{camera}} + \mathcal{L}_{\text{depth}} + \mathcal{L}_{\text{nmap}}, \tag{1}$$

where $\theta$ is the LoRA parameters, $\mathcal{L}_{\text{camera}}$, $\mathcal{L}_{\text{depth}}$ and $\mathcal{L}_{\text{nmap}}$ denote the camera pose loss, the depth loss, and the point map loss, respectively. In the following text, we simply use "VGGT" to refer to this fine-tuned VGGT.

**Generation prior of TRELLIS** TRELLIS Xiang et al. (2024) is a SOTA 3D generative model that provides a strong generation prior. It proposes a novel representation called Structured LATent (SLAT) that combines a sparse 3D grid with dense visual features extracted from a powerful vision foundation model, which captures both geometric (structure) and textural (appearance) information and enables decoding into multiple 3D representations. We choose TRELLIS as the 3D generator in our framework because it has shown great potential in 3D object generation He et al. (2025); Li et al. (2025b) and inspired many works in downstream applications Yang et al. (2025); Cao et al. (2025); Wu et al. (2024c). It employs a coarse-to-fine two-stage generation pipeline: generating the sparse structure (SS), represented as sparse voxels $\{p_i\}_i^L$, via SS Flow and then predicting structured latents

(SLAT) for active SS voxels, represented as $X = \{(p_i, x_i)\}_i^V$, via SLAT Flow, where $p_i$, $x_i$, and $V$ denotes the voxel position, the latent vector, and the number of voxels, respectively. The generation in both stages adopts rectified flow transformers Liu et al. (2022) with DINO-encoded image features as conditions. The result of SLAT Flow is then decoded into 3D outputs represented by radiance fields (RF), 3D Gaussians (3DGS), or meshes, *i.e.*, $O = \text{Dec}(x)$. Modeling the backward process as a time-dependent vector field $\boldsymbol{v}(x, t) = \nabla_t(x)$, the transformers $\boldsymbol{v}_\theta$ in both stages are trained by minimizing the conditional flow matching (CFM) objective Lipman et al. (2023):

$$\mathcal{L}_{\text{CFM}}(\theta) = \mathbb{E}_{t, x_0, \epsilon} \| \boldsymbol{v}_\theta(x, t) - (\epsilon - x_0) \|_2^2. \tag{2}$$

**Overview** Our ReconViaGen framework conducts reconstruction and generation simultaneously and utilizes the two priors in a complementary fashion. It builds upon TRELLIS to generate complete 3D outputs with strong generation priors to plausibly hallucinate invisible portions to compensate for the limitation of reconstruction. The proposed ReconViaGen adopts a coarse-to-fine reconstruction pipeline. As shown in Fig. 2, in the first stage, we use a pre-trained VGGT to provide reconstruction-based multi-view conditions at both the global and local levels. In the next stage, we respectively feed the global geometry and local per-view conditions into the SS and SLAT Flow transformers, for multi-view-aware generation. Finally, we further refine the estimated camera poses from VGGT using the generation and introduce pixel-level alignment constraints only in the inference stage for reconstructions highly consistent with input views in detailed geometry and textures.

## 3.2 RECONSTRUCTION-BASED CONDITIONING

We first introduce reconstruction priors in VGGT to provide strong multi-view-aware conditions for the coarse and detailed shape and texture generation of TRELLIS.

**Global Geometry Condition** VGGT learns a strong reconstruction prior to encode explicit 3D lifting information into multi-view image features. Therefore, we first aggregate VGGT features $\phi_{\text{vggt}}$ into a global geometry representation, serving as SS Flow conditions to generate more accurate coarse structures. Note that we did not use explicit reconstruction results like point clouds because VGGT features convey richer information including camera poses, depth, point maps, and tracking. A fixed-length token list $T_g$ is aggregated from $\phi_{\text{vggt}}$ via a proposed Condition Net design shown in Fig. 2. Starting from a randomly initialized learnable token list $T_{\text{init}}$, four transformer cross-attention blocks progressively fuse layer-wise features of $\phi_{\text{vggt}}$ with the initial token list and produce $T_g$. Formulated as:

$$T^{i+1} = \text{CrossAttn}\big(Q(T^i), K(\phi_{\text{vggt}}), V(\phi_{\text{vggt}})\big), \ \ i \in \{0, 1, 2, 3\}, \tag{3}$$

where $T^0$ is initialized with $T_{\text{init}}$, $T^3$ is the final output $T_g$, $Q(\cdot)$, $K(\cdot)$, and $V(\cdot)$ are linear layers respectively for query, key, and value projection, and $\phi_{\text{vggt}}$ is the VGGT features that concatenate all views on the token dimension. At the training stage of SS Flow, we freeze the VGGT layers and train the Condition Net together with DiT.

**Local Per-View Condition** A single token list condition can provide limited fine-grained information for geometry and texture generation in detail. We further adopt the Condition Net design to provide local per-view tokens as SLAT Flow conditions for fine-grained generation in both geometric and texture details. A random token list is initialized for each view and fed into the Condition Net to produce a view-specific token list $P_k$, $k \in [1, N]$:

$$P_k^{i+1} = \text{CrossAttn}\big(Q(P_k^i), K(\phi_k^{\text{vggt}}), V(\phi_k^{\text{vggt}})\big), \ \ i \in \{0, 1, 2, 3\} \text{ and } k \in \{n\}_{n=1}^N, \tag{4}$$

where $\phi_k^{\text{vggt}}$ is VGGT features of the $k$-th view. The set of $\{P_k\}_{k=1}^N$ is sent into SLAT diffusion transformers offering per-view object appearance guidance for fine-grained generation.

## 3.3 COARSE-TO-FINE GENERATION

The overall generation process consists of three stages: (i) coarse structure generation via SS Flow with global geometry condition; (ii) fine detail generation via SLAT Flow with local per-view condition; (iii) rendering-aware pixel-aligned refinement at the inference stage only.

**Reconstruction-conditioned Flow** To integrate the reconstruction prior into generation, the two stages of SS and SLAT Flow in TRELLIS take the global geometry condition $T_g$ and local per-view conditions $\{P_k\}_{k=1}^N$ for coarse and fine diffusion guidance, respectively. In the first stage, we simply compute the cross-attention between the condition $T_g$ and the noisy SS latent in each SS DiT block. In the second stage, as illustrated in Fig. 2, we encourage the cross-attention between the noisy SLAT and each view's condition $P_k$ and conduct a weighted fusion in each SLAT DiT block, which can be formulated as:

$$y_{j+1} = \sum_{k=1}^N \text{CrossAttn}\big(Q(y'_j), K(P_k), V(P_k)\big) \cdot w_k, \;\; j \in \{m\}_{m=1}^M, \tag{5}$$

where $M$ is the number of SLAT DiT blocks, $y'_j$ is the self-attention layer output of the noisy SLAT input $y_j$, and $w_k \in (0, 1)$ is the fusion weight computed via an MLP taking the cross-attention result as input. After the first two stages, the 3D generator can generate multi-view-aware geometry and texture at both the global and local levels.

**Rendering-aware Velocity Compensation** To further encourage pixel-aligned consistency between generation results and input views, we develop a rendering-aware velocity compensation to constrain the diffusion trajectory according to inputs. In doing so, we first estimate camera pose with VGGT using the generation results from the second stage, with detailed implementation details included in the appendix. Inspired by the explicit normal regularization used in Hi3DGen Ye et al. (2025) to improve the input-output consistency, when $t < 0.5$, we decode the SLAT into $O_t$ (*e.g.* a textured mesh) and conduct rendering for alignment. The SLAT Flow process initializes and updates a large number of noisy latents for all voxels simultaneously, which results in a challenging collaborative optimization problem. To solve this issue, we novelly propose a mechanism called Rendering-aware Velocity Compensation (RVC) to correct the predicted $v$ for a more accurate generation consistent with input views. Specifically, we render images for $O_t$ from the refined camera pose estimations $C$ and calculate the difference between the rendered images and input images as:

$$\mathcal{L}_{\text{RVC}}(v_t) = \mathcal{L}_{\text{SSIM}} + \mathcal{L}_{\text{LPIPS}} + \mathcal{L}_{\text{DreamSim}}, \tag{6}$$

where $\mathcal{L}_{\text{SSIM}}$, $\mathcal{L}_{\text{LPIPS}}$, and $\mathcal{L}_{\text{DreamSim}}$ are SSIM Wang et al. (2004), LPIPS Zhang et al. (2018), and DreamSim losses Fu et al. (2023) (inspired by the practice in V2M4 Chen et al. (2025)), responsible for measuring the structural, perceptual, and semantic similarity, respectively. To exclude the influence of inaccurate pose estimation, we discard the losses corresponding to some images if their corresponding losses are higher than 0.8. By minimizing $\mathcal{L}_{\text{RVC}}$, we iteratively correct the predicted velocity in each SLAT denoising step with a compensation term $\Delta v$, derived as:

$$\Delta v_t = \frac{\partial \mathcal{L}}{\partial \hat{x}_0} \frac{\partial \hat{x}_0}{\partial v_t} = -t \frac{\partial \mathcal{L}}{\partial \hat{x}_0}, \tag{7}$$

where $\mathcal{L}$ represents $\mathcal{L}_{\text{RVC}}$ for simplicity and $\hat{x}_0$ is the predicted target SLAT at current timestep $t$, computed as $\hat{x}_0 = x_t - t \cdot v_t$. The noisy SLAT of next step $x_{t_{\text{prev}}}$ can be updated as:

$$x_{t_{\text{prev}}} = x_t - (t - t_{\text{prev}})(v + \alpha \cdot \Delta v), \tag{8}$$

where $\alpha$ is a pre-defined hyperparameter that controls the extent of the compensation. In this way, the input images serve as a strong explicit guidance to find a denoising trajectory for each local SLAT vector, which leads to more accurate 3D results consistent with all input images in detail.

## 4 EXPERIMENTS

### 4.1 EXPERIMENT SETUP

**Datasets** For LoRA fine-tuning of the VGGT aggregator and Trellis sparse structure transformer, we employ 390k 3D data from the Objaverse dataset Deitke et al. (2024), a large-scale 3D object dataset that provides a rich variety of shapes and textures, with 60 views rendered per object for fine-tuning. For each object mesh, we render 150 view images in a resolution of $512 \times 512$ under uniform lighting conditions following TRELLIS Xiang et al. (2024). For evaluation, we selected two benchmark datasets to thoroughly assess the performance of our model: (i) Dora-Bench Chen et al. (2024), a benchmark organized based on 4 levels of complexity, combining 3D data selected

from the Objaverse Deitke et al. (2023), ABO Collins et al. (2022), and GSO Downs et al. (2022) dataset; and (ii) OmniObject3D, a large-vocabulary 3D object dataset containing 6,000 high-quality textured meshes scanned from real-world objects, covering 190 daily categories. We randomly sample 300 objects from Dora-Bench and 200 objects covering 20 categories from OmniObject3D. We follow He et al. (2024) to render 24 views at different elevations, and randomly chose 4 of them as multi-view input for evaluation on OmniObject3D. On Dora-Bench, we follow the camera trajectory of TRELLIS Xiang et al. (2024) to render 40 views and choose 4 views (No.0, 9, 19, and 29) with a uniform interval to adapt to the setting of some baseline methods (LGM Tang et al. (2025) and InstantMesh Xu et al. (2024b)).

**Evaluation Metrics** We employ PSNR, SSIM, and LPIPS to evaluate the accuracy of synthesized novel views from 3D outputs, Chamfer Distance (CD) and F-score to evaluate the generated geometry accuracy and completeness. PSNR, SSIM, and LPIPS are evaluated on novel views of images rendered at the resolution of $512 \times 512$. CD and F-score are evaluated by sampling 100k points from the 3D outputs (using the center positions for 3D Gaussian outputs), with all object points normalized to the range of $[-1, 1]^3$. When calculating the F-score, the radius $r$ is set to 0.1.

**Baseline Methods** Baseline methods for comparisons include (i) TRELLIS-S Xiang et al. (2024): generates 3D meshes from multi-view images using TRELLIS in the stochastic mode, which randomly chooses one input view to condition each step of denoising; (ii) TRELLIS-M Xiang et al. (2024): TRELLIS in the multidiffusion mode, which computes the average denoised results conditioned on all input views; (iii) Hunyuan3D-2.0-mv Zhao et al. (2025): concatenate DINO features of input images from fixed viewpoints as conditions to generate meshes[1]; (iv) InstantMesh Xu et al. (2024b): predicts Triplane for mesh outputs from multiple images with fixed viewpoints; (v) LGM Tang et al. (2025): predicts pixel-aligned 3D Gaussians from multiple images with fixed viewpoints; (vi) LucidFusion He et al. (2024): predicts relative coordinate maps for 3D Gaussian outputs; (vii) VGGT Wang et al. (2025a): reconstructs the point cloud from multi-view inputs in a feed-forward manner. We compare our methods with a wide range of existing SOTA baseline methods: (a) 3D generation models {i, ii, iii}; (b) large reconstruction models with known camera poses {iv, v}; (c) pose-free large reconstruction models with 3DGS or point cloud outputs {vi, vii}. For 3D generation models {i, ii, iii}, we use the same approach as Camera Pose Estimation in fine detail reconstruction to align the generated 3D models to the ground-truth models. Besides, we also compare with closed-source commercial 3D generation models like Hunyuan3D-2.5 and Meshy-5 on in-the-wild testing.

**Implementation Details** For LoRA fine-tuning of VGGT aggregator and TRELLIS transformer, we set the rank as 64, the alpha parameter for LoRA scaling as 128, and the dropout probability for LoRA layers as 0. We only apply the adapter to qkv mapping layer and the projectors of each attention layer. During fine-tuning VGGT aggregator, we randomly sample $1 \sim 4$ views from 150 images and use the AdamW optimizer with a fixed learning rate of $1 \times 10^{-4}$. For the fine-tuning of SS Flow and Slat-Flow transformer, we build upon TRELLIS Xiang et al. (2024), incorporating classifier-free guidance (CFG) with a drop rate of 0.3 and an AdamW optimizer with a fixed learning rate of $1 \times 10^{-4}$. We fine-tune the SS-Flow transformer using 8 NVIDIA A800 GPUs (80GB memory) for 40k steps with a batch size of 192. Differently, we finetune the SLat-Flow transformer with a batch size of 128. During inference, we set the CFG strengths in SS generation and SLAT generation to 7.5 and 3.0, and use 30 and 12 sampling steps to achieve optimal results. The $\alpha$ in rendering-aware velocity compensation is set to 0.1 in our practice.

## 4.2 EXPERIMENT RESULTS

**Quantitative Results** We present the quantitative comparisons between our ReconViaGen and other baseline methods in Tab. 1 for evaluation on the Dora-bench and OminiObject3D dataset. The proposed method achieves consistently superior performance to other methods on both image-reconstruction consistency (PSNR, SSIM, and LPIPS), geometry accuracy (CD), and shape completeness (F-score). Impressively, our ReconViaGen seamlessly integrates the generation and reconstruction priors from TRELLIS Xiang et al. (2024) and VGGT Wang et al. (2025a), whose performance surpasses both of them. Note that VGGT performs better on Dora-bench than on OmniObject3D because uniformly-distributed views can capture richer visual cues than random views.

---

[1]The fresh version, Hunyuan3D-2.5, has not been open-sourced, which is unsuitable for large-scale evaluation on benchmarks, so we use the open-sourced version, Hunyuan3D-2.0.

Table 1: Evaluation on the Dora-bench and OmniObject3D dataset. Best results are in **bold**.

| Method | Dora-bench | | | | | OmniObject3D | | | | |
|---|---|---|---|---|---|---|---|---|---|---|
| | PSNR↑ | SSIM↑ | LPIPS↓ | CD↓ | F-score↑ | PSNR↑ | SSIM↑ | LPIPS↓ | CD↓ | F-score↑ |
| VGGT Wang et al. (2025a) | - | - | - | 0.112 | 0.921 | - | - | - | 0.091 | 0.900 |
| TRELLIS-S Xiang et al. (2024) | 16.562 | 0.876 | 0.103 | 0.176 | 0.807 | 16.021 | 0.771 | 0.264 | 0.102 | 0.906 |
| TRELLIS-M Xiang et al. (2024) | 16.706 | 0.882 | 0.111 | 0.144 | 0.843 | 16.861 | 0.790 | 0.242 | 0.072 | 0.932 |
| Hunyuan3D-2.0-mv Zhao et al. (2025) | 20.221 | 0.896 | 0.093 | 0.094 | 0.937 | 16.665 | 0.813 | 0.165 | 0.124 | 0.871 |
| LGM Tang et al. (2025) | 17.877 | 0.869 | 0.186 | 0.121 | 0.839 | 16.361 | 0.791 | 0.193 | 0.136 | 0.842 |
| InstantMesh Xu et al. (2024b) | 18.922 | 0.870 | 0.120 | 0.110 | 0.865 | 17.499 | 0.818 | 0.145 | 0.094 | 0.907 |
| LucidFusion He et al. (2024) | 16.509 | 0.835 | 0.144 | 0.131 | 0.831 | 16.254 | 0.771 | 0.144 | 0.114 | 0.868 |
| **ReconViaGen (Ours)** | **22.632** | **0.911** | **0.090** | **0.090** | **0.953** | **19.767** | **0.847** | **0.141** | **0.059** | **0.959** |

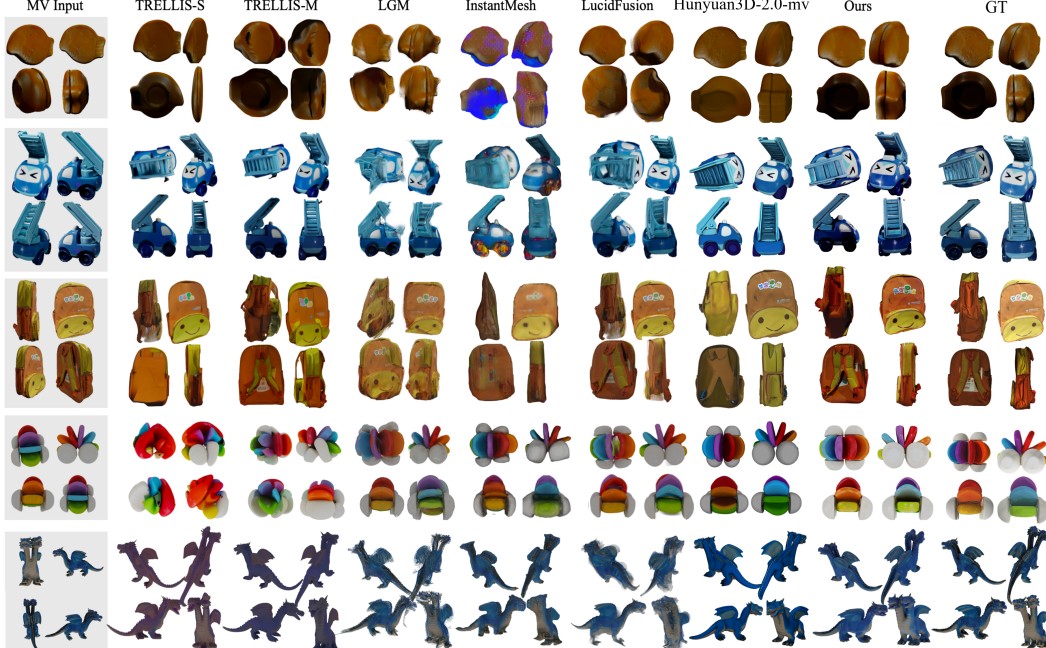

Figure 3: Reconstruction result comparisons between our ReconViaGen and other baseline methods on samples from the Dora-bench and OmniObject3D datasets. Zoom in for better visualization.

Besides, our method also gets better results than previous SOTA pose-free multi-view reconstruction methods that integrate regression-based generation priors by a large margin, especially on PSNR, CD, and F-score, which validates the superiority of ReconViaGen. On the settings of more input views, we separately evaluate VGGT that can accept an arbitrary number of inputs for comparison, which is included in the appendix. We also present the camera pose estimation accuracy in the appendix.

**Qualitative Results** We further present extensive qualitative comparisons to demonstrate the superiority of ReconViaGen. We first select some examples from the OmniObject3D and Dora-bench dataset for visualization, as shown in Fig. 3. The reconstruction results of ReconViaGen have the most accurate geometry and textures compared to other methods. We further evaluate several baseline methods on in-the-wild multi-view images. As shown in Fig. 4, ReconViaGen exhibits strong robustness even in comparison with the multi-view version of closed-source commercial 3D generation models like Hunyuan3D-2.5 and Meshy-5. More qualitative results are included in the appendix.

Table 2: Quantitative ablation results on the Dora-bench dataset.

| | GGC | PVC | RVC | PSNR↑ | SSIM↑ | LPIPS↓ | CD↓ | F-score↑ |
|---|---|---|---|---|---|---|---|---|
| (a) | ✗ | ✗ | ✗ | 16.706 | 0.882 | 0.111 | 0.144 | 0.843 |
| (b) | ✓ | ✗ | ✗ | 20.462 | 0.894 | 0.102 | 0.093 | 0.941 |
| (c) | ✓ | ✓ | ✗ | 21.045 | 0.905 | 0.093 | 0.093 | 0.937 |
| (d) | ✓ | ✓ | ✓ | **22.632** | **0.911** | **0.090** | **0.090** | **0.953** |

MV Input   TRELLIS-S   TRELLIS-M   Meshy-5-mv   Hunyuan3D-2.5-mv   Ours

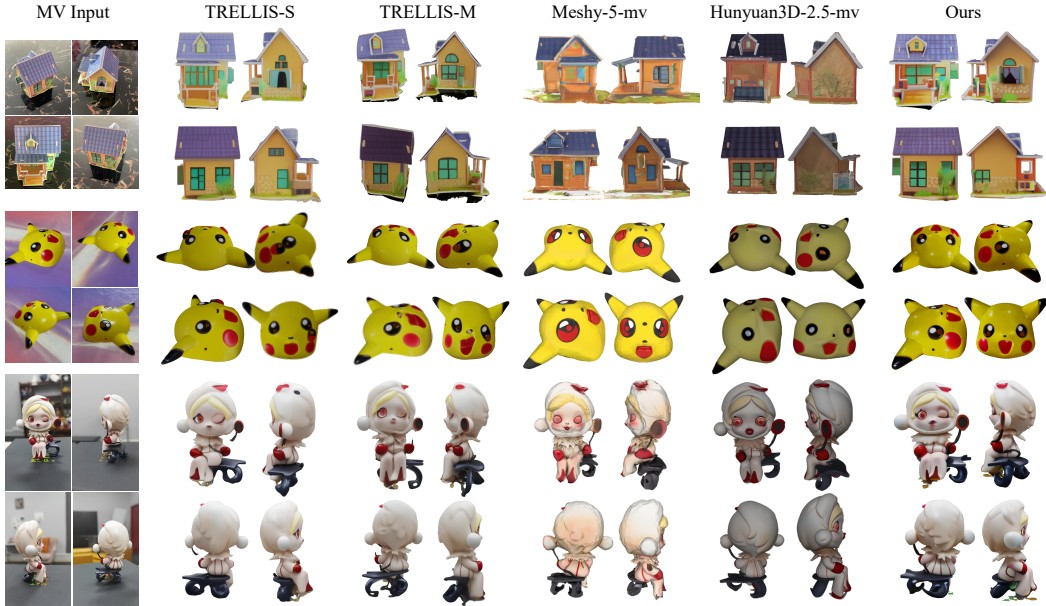

Figure 4: Reconstruction results on in-the-wild samples. Note that commercial 3D generators require input images from orthogonal viewpoints, while ours can accept views from arbitrary camera poses for robust outputs. Zoom in for better visualization in detail.

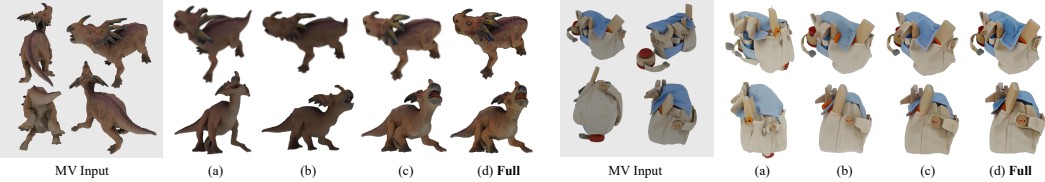

MV Input   (a)   (b)   (c)   (d) **Full**       MV Input   (a)   (b)   (c)   (d) **Full**

Figure 5: Qualitative comparisons for different variants of ReconViaGen for ablative study. Zoom in for better visualization in detail.

## 4.3 ABLATION STUDY

The proposed ReconViaGen framework comprises three novel designs to integrate reconstruction priors into the diffusion-based 3D generation: (i) the global geometry condition (GGC); (ii) the per-view condition (PVC); and (iii) the rendering-aware velocity compensation (RVC). We conduct ablation studies to validate the individual effectiveness of each component. On the Dora-bench dataset, we start from a basic TRELLIS-M baseline (ReconViaGen without all designs, Tab. 2a) and progressively add one component, leading to 3 variants (b,c,d). As shown in Tab. 2, integrating GGC, which strongly improves the prediction accuracy of coarse structure, brings a large performance gain on almost all metrics. Further integrating PVC can lead to extra improvement, especially on PSNR, which proves the effectiveness in improving local per-view alignment. Finally, adopting RVC, though in the inference stage only, brings additional increments in both shape completeness and fine-grained accuracy in geometry and texture. Qualitative comparisons in Fig. 5 visualize the positive effect of each component: global geometry conditioning greatly corrects the global shape, per-view conditioning produces local details in geometry and texture of high consistency with each view, and rendering-aware velocity compensating impressively refines the fine-grained appearance, leading to high-quality results.

Since ReconViaGen can take an arbitrary number of images, a natural question is how reconstruction quality scales with the number of images. To investigate this, we conducted an ablation study varying the number of input views on Dora-Bench, with results summarized in Tab. 3. We observe that reconstruction performance consistently improves as more images are provided. However, the marginal gains gradually diminish, indicating a saturation effect when the number of views becomes large. The visualization results are shown in Fig. 6, which also shows that ReconViaGen can process

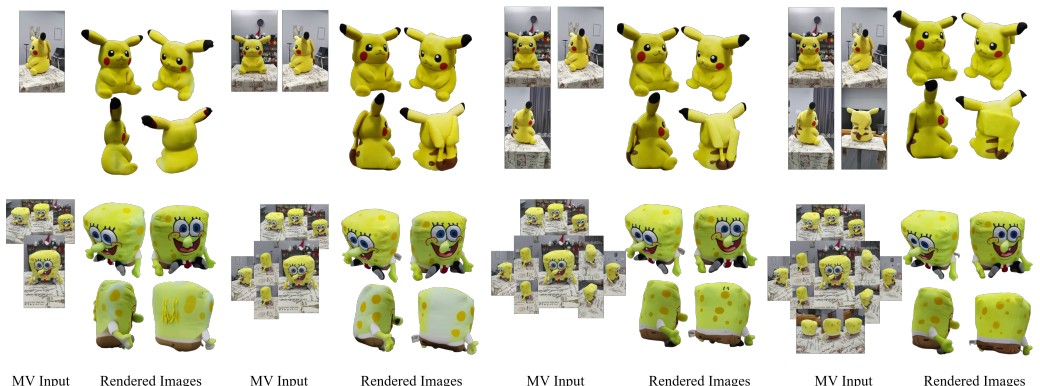

MV Input    Rendered Images    MV Input    Rendered Images    MV Input    Rendered Images    MV Input    Rendered Images

Figure 6: Qualitative comparisons for different numbers of input images with ReconViaGen. Zoom in for better visualization in detail.

any number of input images from any viewpoint. More ablation results on the detailed designs, including the choice of condition form for SS and SLAT Flow, can be seen in the appendix.

Table 3: Quantitative ablation results of the number of input images on the Dora-bench dataset.

| Number of Images | PSNR↑ | SSIM↑ | LPIPS↓ | CD↓ | F-score↑ |
|---|---|---|---|---|---|
| 1 | 18.438 | 0.887 | 0.106 | 0.135 | 0.838 |
| 2 | 19.568 | 0.894 | 0.099 | 0.131 | 0.867 |
| 4 | 22.632 | 0.911 | 0.090 | 0.090 | 0.953 |
| 6 | 22.823 | 0.912 | **0.089** | 0.084 | 0.958 |
| 8 | **23.067** | **0.914** | 0.090 | **0.081** | **0.961** |

## 5 CONCLUSION

In this paper, we have presented ReconViaGen, a novel coarse-to-fine framework that effectively integrates strong reconstruction priors with diffusion-based 3D generative priors for accurate and complete multi-view 3D object reconstruction. We first analyze the inherent reasons leading to the challenge of leveraging diffusion-based 3D generative priors into reconstruction: insufficient cross-view correlation modeling and stochastic denoising process with weak constraint from input images. Therefore, we effectively use powerful reconstruction priors with three novelly designed mechanisms to enhance the multi-view correlation awareness in 3D diffusion learning and establish strong constraints for a reliable denoising process. Extensive experiments have demonstrated that ReconViaGen achieves SOTA performance in both global shape accuracy and completeness as well as local details in geometry and textures. As future work, with the development of 3D reconstruction and 3D generation, stronger reconstruction or generation priors can be integrated into our framework to further improve reconstruction quality via generation.

## 6 ACKNOWLEDGMENTS

The work was supported in part by Guangdong Provincial Outstanding Youth Fund with No. 2023B1515020055, the Shenzhen Outstanding Talents Training Fund 202002, the NSFC with Grant No. 62293482, the Guangdong Research Projects No. 2017ZT07X152 and No. 2019CX01X104, the Guangdong Provincial Key Laboratory of Future Networks of Intelligence (Grant No. 2022B1212010001), and the Shenzhen Key Laboratory of Big Data and Artificial Intelligence (Grant No. SYSPG20241211173853027), the Guangdong Province Radio Science Data Center.

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
