# Supplementary Material for ReconViaGen: Towards Accurate Multi-view 3D Object Reconstruction via Generation

**Jiahao Chang**[1,2*]   **Chongjie Ye**[2,1*]   **Yushuang Wu**[2]   **Yuantao Chen**[1]
**Yidan Zhang**[1]   **Zhongjin Luo**[2]   **Chenghong Li**[2]   **Yihao Zhi**[1]   **Xiaoguang Han**[1,2,3†]
[1]School of Science and Engineering, The Chinese University of Hong Kong, Shenzhen
[2]Shenzhen Future Network of Intelligence Institute
[3]Guangdong Provincial Key Laboratory of Future Networks of Intelligence

## 1 Appendix

### A.1 Details on camera pose estimation

For better alignment with the input images, we register them into the TRELLIS generation space. Specifically, we first render 30 images from randomly sampled camera views on a sphere, concatenate them with the input images, and feed them into VGGT for pose estimation. Since the camera poses of the rendered views are known, we can recover coarse camera poses for the input images in the TRELLIS space. While VGGT provides robust pose predictions, they remain insufficiently accurate for constructing pixel-level rendering constraints.

To refine the results, we render images and depth maps using the coarse poses, then apply an image matching method to establish 2D-2D correspondences between rendered and input images. Leveraging the depth maps and camera parameters of rendered views, we further obtain 2D-3D correspondences between each input image and the generated object. By aggregating multi-view correspondences, we solve for refined camera poses $C$ using a PnP Lepetit et al. (2009) solver with RANSAC Fischler & Bolles (1981). This image-matching-based refinement effectively corrects the initial pose predictions from TRELLIS's generative priors, yielding higher accuracy. The refined poses enable pixel-wise constraints from the input views, thereby supporting finer detail alignment in generation.

### A.2 Evaluation with more input images

Table 1: Evaluation with more input images on the Dora-bench dataset. Best results are in **bold**.

| Method | Uniform (PSNR↑ / LPIPS↓) | | | Limited View (PSNR↑ / LPIPS↓) | | |
|---|---|---|---|---|---|---|
| | 6 views | 8 views | 10 views | 6 views | 8 views | 10 views |
| Object VGGT + 3DGS | 18.476/0.123 | 19.890/0.109 | 21.363/0.102 | 16.498/0.139 | 16.774/0.135 | 17.121/0.133 |
| **ReconViaGen** (Ours) | **22.823/0.089** | **23.067/0.090** | **23.193/0.087** | **21.427/0.098** | **21.782/0.099** | **21.866/0.103** |

To thoroughly evaluate and validate the effectiveness of ReconViaGen, we compare it against 3DGS reconstruction initialized with point clouds and camera poses from object VGGT (denoted as object VGGT + 3DGS) on the Dora-Bench dataset. We conduct experiments under two input scenarios: uniformly and limited-view sampled views. As shown in Tab. 1, ReconViaGen consistently outperforms object VGGT+3DGS at 6/8/10 input views, regardless of the sampling strategy. This advantage arises because the generative prior in ReconViaGen plays a crucial role in completing invisible regions of the object.

Table 2: Evaluation of camera pose estimation on the Dora-bench dataset. Best results are in bold.

| Method | RRE↓ | Acc.@15° ↑ | Acc.@30° ↑ | TE↓ |
|---|---|---|---|---|
| VGGT Wang et al. (2025) | 8.575 | 90.67 | 92.00 | 0.066 |
| Object VGGT | **7.257** | 93.44 | 94.11 | 0.055 |
| Ours | 7.925 | **93.89** | **96.11** | **0.046** |

## A.3 EVALUATION OF CAMERA POSE ESTIMATION

To assess the performance of our finetuned object VGGT and the effectiveness of our proposed camera pose estimation strategy, we evaluate pose prediction quality on the Dora-Bench dataset. We adopt both rotation and translation metrics: relative rotation error (RRE, in degrees), the proportion of RRE values below $15°$ and $30°$, and translation error (TE), measured as the distance between predicted and ground-truth camera centers. For evaluation, we use four input images and transform both predicted and ground-truth poses into the coordinate system of the first image, which is excluded from the metric computation. To address translation scale ambiguity, we compute relative translations between views for both predictions and ground truth and normalize them by their respective mean L2-norm. As reported in Tab. 2, the finetuned object VGGT achieves clear improvements over the original VGGT. Our method further delivers the best overall performance, as the generative prior effectively "densifies" sparse views. However, our RRE is slightly higher than that of object VGGT, likely due to minor discrepancies between the generated 3D model and the ground-truth geometry.

## A.4 ABLATION STUDY ON THE FORM OF CONDITION

Table 3: Quantitative ablation results of condition at SS Flow on the Dora-bench dataset.

| | Form of Condition | PSNR↑ | SSIM↑ | LPIPS↓ | CD↓ | F-score↑ |
|---|---|---|---|---|---|---|
| (i) | Feature Volume | 16.229 | 0.858 | 0.126 | 0.172 | 0.814 |
| (ii) | Concatenation | 19.749 | 0.871 | 0.137 | 0.121 | 0.873 |
| (iii) | PVC | 19.878 | 0.882 | 0.135 | 0.120 | 0.870 |
| (iv) | GGC | **20.462** | **0.894** | **0.102** | **0.093** | **0.941** |

For SS Flow, as described in the method section, we explored several strategies to leverage VGGT features for sparse structure generation on Dora-Bench: (i) Downsampling the point cloud from VGGT to a $64^3$ resolution occupancy volume, projecting DINO features from each view into the volume, and averaging them to form a feature-volume condition; (ii) Fusing VGGT features with DINO features for each view through several linear layers, then concatenating all input-view tokens as conditions; (iii) adopting the same local per-view condition (PVC) used in our SLAT Flow; (iv) employing the proposed global geometry condition (GGC). For fair comparison, we use the original SLAT Flow in TRELLIS and train all models for 40k steps. As shown in Tab. 3, our GGC achieves the best performance among all strategies. We attribute this to the limitations of the alternative designs: for (i), inaccurate predicted poses or point clouds lead to erroneous projections, introducing noise into the condition and harming generation; for (ii) and (iii), view-level features are not effectively aggregated, resulting in redundancy and making the model overly dependent on the accuracy of VGGT outputs.

Table 4: Quantitative ablation results of condition at SLAT Flow on the Dora-bench dataset.

| | Form of Condition | PSNR↑ | SSIM↑ | LPIPS↓ | CD↓ | F-score↑ |
|---|---|---|---|---|---|---|
| (i) | GGC | 17.784 | 0.858 | 0.120 | 0.097 | 0.939 |
| (ii) | PVC | **22.632** | **0.911** | **0.090** | **0.090** | **0.953** |

For SLAT Flow, we conduct an ablation study on Dora-Bench with two conditioning strategies: (i) the same global geometry condition (GGC) used in SS Flow, and (ii) the local per-view condition (PVC). For fairness, we pair both variants with SS Flow conditioned on GGC and train all models for 40k steps. As shown in Tab. 4, PVC substantially outperforms GGC in SLAT Flow. We attribute this to the information compression in GGC, which leads to a loss of fine-grained details in the

---

*Equal contribution.

†Corresponding Author.

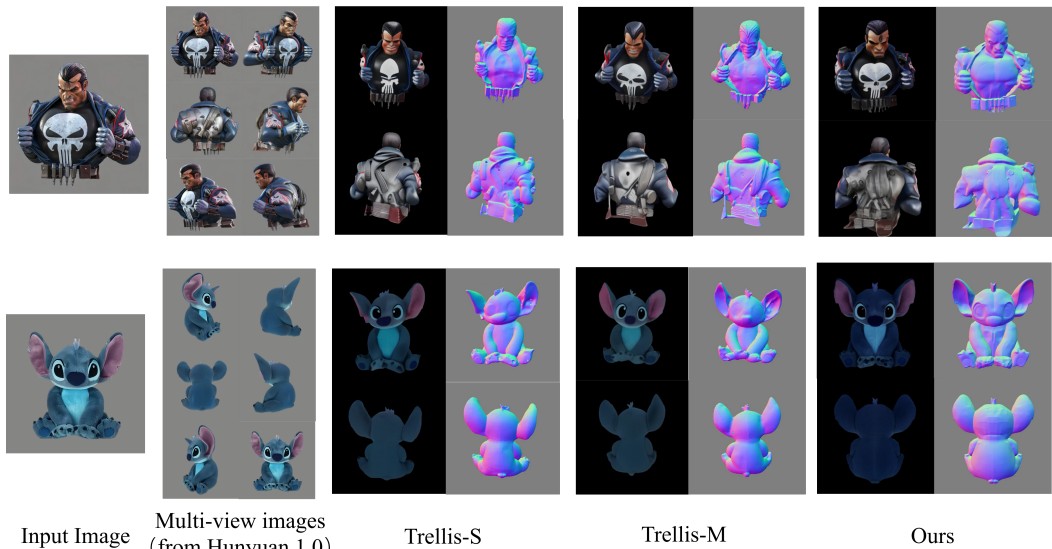

| Input Image | Multi-view images (from Hunyuan 1.0) | Trellis-S | Trellis-M | Ours |

Figure 1: Reconstruction result comparisons between TRELLIS-M, TRELLIS-S, and our ReconViaGen on samples produced by the multi-view image generator.

condition and degrades performance. This observation also explains why we adopt PVC instead of GGC for SLAT Flow.

The following description provides an interpretability of GGC and PVC. GGC performs global multi-view aggregation, which strengthens overall structural consistency—this is exactly what SS needs, since SS focuses on coarse geometry where global cues dominate and multi-view compression is acceptable. In contrast, SLAT requires fine-grained appearance and local geometry refinement on a fixed coarse shape. PVC's per-view feature interaction preserves high-frequency cues that GGC would smooth out. Therefore, GGC naturally aligns with global structure formation, while PVC is better suited for detail-aware refinement.

## A.5 RECONSTRUCTION ON GENERATED MULTI-VIEW IMAGES OR VIDEOS

Table 5: Quantitative comparison of generated multi-view images on the Dora-bench dataset.

| Method | PSNR↑ | SSIM↑ | LPIPS↓ | CD↓ | F-score↑ |
|---|---|---|---|---|---|
| ours (generated 6-view) | 14.379 | 0.808 | 0.226 | 0.190 | 0.723 |
| ours (single-view) | 18.438 | 0.887 | 0.106 | 0.135 | 0.838 |
| ours (real 6-view) | **22.823** | **0.912** | **0.089** | **0.084** | **0.958** |

Given the growing interest in multi-view image generation using large image and video generative models, we further evaluate the robustness of our approach on such generated data. These multi-view images are hallucinated from a single view and often suffer from cross-view inconsistencies in fine details. Specifically, we generate 6-view samples using the open-sourced multi-view generator Hunyuan3D-1.0 Yang et al. (2024). The quantitative comparisons on Dora-Bench are summarized in Tab. 5. In our observations, the results of generated 6 views as input are much worse than those of single-view and real 6-view input due to much more severe inconsistency in multi-view generation. However, when the inconsistency in generated multi-view images is not severe, visualizations in Fig. 1 show that ReconViaGen exhibits strong robustness to moderate cross-view inconsistency. Please refer to the supplementary video for additional results on generated videos.

## A.6 MORE RECONSTRUCTION RESULTS

We further showcase our method on in-the-wild data, including not only multiple objects but also scenes, even from generated dynamic object videos. For scene reconstruction, we segment individual objects, reconstruct them separately, and then register the reconstructed 3D objects back into

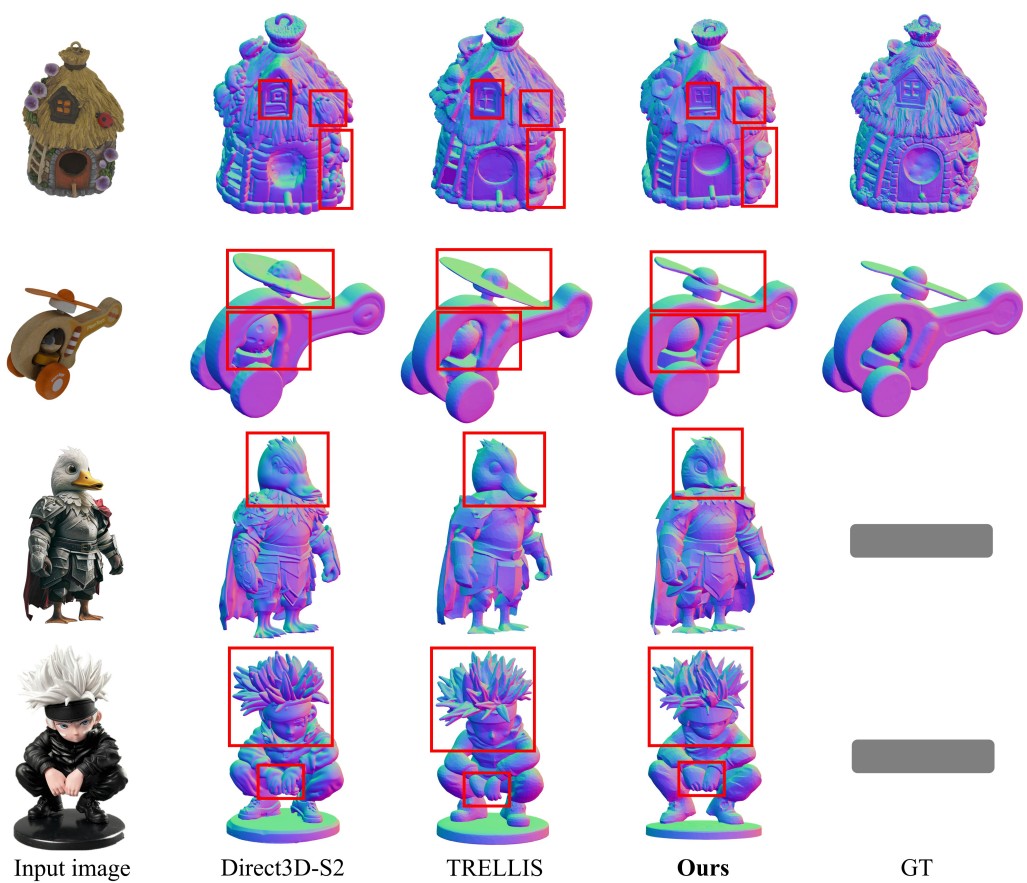

Figure 2: Qualitative result of our ReconViaGen and other baseline methods with single-view input on samples from the Dora-bench and in-the-wild scenarios. Since in-the-wild cases lack corresponding ground truth meshes, these ground truth meshes are not displayed in the figure. Zoom in for better visualization.

the scene using our predicted camera poses. Please refer to the supplementary video for qualitative results.

## A.7 COMPARISON EXPERIMENT OF SINGLE-VIEW INPUT

Table 6: Quantitative results of single view input on the Dora-bench dataset.

| Method | PSNR↑ | SSIM↑ | LPIPS↓ | CD↓ | F-score↑ |
|---|---|---|---|---|---|
| TRELLIS Xiang et al. (2024) | 15.264 | 0.858 | 0.182 | 0.162 | 0.781 |
| Direct3D-S2 Wu et al. (2025) | - | - | - | 0.165 | 0.805 |
| **Ours** | **18.438** | **0.887** | **0.106** | **0.135** | **0.838** |

To further demonstrate the superiority of our ReconViaGen, we conduct comparison experiments with single-view input. We select TRELLIS Xiang et al. (2024) and Direct3D-S2 Wu et al. (2025) as representative single-view generation baselines for comparison. For evaluation on the Dora-Bench dataset, we use a single randomly selected view as input. Quantitative comparisons between our ReconViaGen and the baseline methods are summarized in Tab. 6. Note that since Direct3D-S2 Wu et al. (2025) outputs geometry without color, we do not report its visual metrics. The quantitative data demonstrate that ReconViaGen consistently achieves superior performance across both visual and geometric evaluation criteria. We further provide extensive qualitative comparisons on both Dora-Bench and in-the-wild scenarios, as shown in Fig. 2. The visual evidence confirms that our

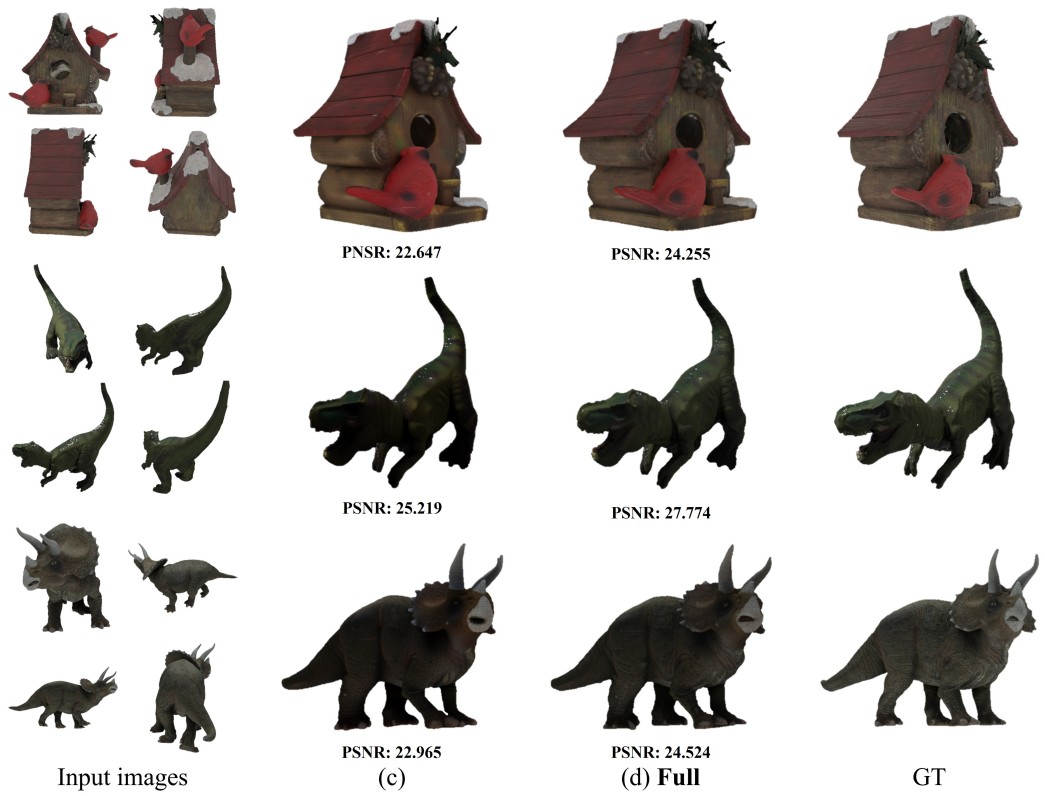

Figure 3: More qualitative comparisons for (c) vs (d) ablation study. Each case is labeled with a corresponding PSNR value. Zoom in for better visualization in detail.

proposed method consistently outperforms the baseline approaches, yielding reconstructed geometry and textures that exhibit the strongest fidelity and alignment with the input view.

## A.8   MORE QUALITATIVE EXAMPLES FOR THE (C) VS (D) ABLATION STUDY

We present more qualitative examples and label the PSNR of each case for the (c) vs (d) ablation study in Fig. 3. The qualitative analysis confirms that the 1.587 dB improvement is primarily due to RVC's ability to correct color and texture drift and enforce high-frequency alignment with the input views.

## A.9   ABLATION STUDY ON RVC HYPERPARAMETERS

Table 7: Ablation study on the extent of RVC $\alpha$ on the Dora-bench dataset.

| $\alpha$ | PSNR↑ | SSIM↑ | LPIPS↓ | CD↓ | F-score↑ |
|---|---|---|---|---|---|
| 1 | 22.475 | 0.911 | 0.091 | 0.091 | 0.941 |
| 0.5 | 22.151 | **0.913** | 0.093 | **0.089** | 0.949 |
| 0.1 | **22.632** | 0.911 | **0.090** | 0.090 | **0.953** |
| 0.05 | 21.321 | 0.910 | 0.089 | 0.091 | **0.953** |
| 0.01 | 21.079 | 0.906 | 0.095 | 0.092 | 0.941 |

The RVC stage in ReconViaGen involves three main hyperparameters: the extent of RVC $\alpha$, timestep $t$, and outlier rejection threshold $t_o$. Then we conduct extensive ablation experiments on these three hyperparameters on the Dora-Bench dataset. As shown in Tab. 7, for $\alpha$, varying $\alpha$ from 1 to 0.01, we observe stable performance across a wide range, with the best results around $\alpha = 0.1$.

Table 8: Ablation study on timestep $t$ in RVC on the Dora-bench dataset.

| $t$ | PSNR↑ | SSIM↑ | LPIPS↓ | CD↓ | F-score↑ | Runtime↓ |
|---|---|---|---|---|---|---|
| 0.7 | 22.149 | 0.907 | 0.090 | 0.094 | 0.938 | 9.2767 |
| 0.6 | 22.477 | 0.912 | 0.091 | 0.094 | 0.940 | 8.2999 |
| 0.5 | **22.632** | 0.911 | 0.090 | 0.090 | **0.953** | 6.8652 |
| 0.4 | 22.270 | 0.909 | 0.089 | **0.089** | 0.944 | 5.7155 |
| 0.3 | 22.057 | **0.915** | **0.088** | 0.091 | 0.937 | **4.7825** |

Table 9: Ablation study on the outlier rejection threshold $t_o$ in RVC on the Dora-bench dataset.

| $t_o$ | PSNR↑ | SSIM↑ | LPIPS↓ | CD↓ | F-score↑ |
|---|---|---|---|---|---|
| 1 | 21.618 | 0.912 | 0.094 | 0.091 | 0.938 |
| 0.9 | 22.075 | 0.906 | 0.090 | 0.091 | 0.946 |
| 0.8 | **22.632** | **0.911** | **0.090** | 0.090 | **0.953** |
| 0.7 | 22.450 | 0.909 | 0.094 | **0.089** | 0.941 |
| 0.6 | 22.312 | 0.906 | 0.093 | 0.094 | 0.944 |

Extremely small values slightly reduce accuracy, but no instability was observed. Importantly, $\alpha$ introduces no runtime overhead, as it only changes the update weight of a vector. As shown in Tab. 8, applying RVC at different timesteps shows that $t = 0.5$ offers the best trade-off between fidelity and cost. As expected, runtime scales linearly with the fraction of steps with RVC (due to decoding/rendering), ranging from 4.8s ($t = 0.3$) to 9.3s ($t = 0.7$). As shown in Tab. 9, for outlier rejection threshold $t_o$, results are insensitive across a broad range. We choose $t_o = 0.8$ as it yields a strong PSNR/F-score. Therefore, $\alpha = 0.1$, $t = 0.5$, and $t_o = 0.8$ is the optimal combination of hyperparameters. Importantly, the best-performing configuration ($\alpha = 0.1, t = 0.5, t_o = 0.8$) is shared across all assets and datasets, and no per-asset tuning is required. This confirms that the final RVC step generalizes well and does not rely on instance-specific adjustments.

## A.10 THE DETAILED LATENCY OF EACH COMPONENT IN RECONVIAGEN

Table 10: Inference time (in seconds) of each component under varying numbers of input views.

| Component \ Number of images | 1 | 3 | 5 | 7 | 9 |
|---|---|---|---|---|---|
| DINO feature + VGGT feature | 0.2157s | 0.4393s | 0.8789s | 1.3788s | 2.0745s |
| GGC | 0.0169s | 0.0175s | 0.0184s | 0.0199s | 0.0224s |
| SS Flow + SS Decoder | 4.4193s | 4.4039s | 4.4105s | 4.4141s | 4.4267s |
| PVC | 0.0017s | 0.0019s | 0.0022s | 0.0023s | 0.0025s |
| SLat Flow + SLat Decoder | 2.7178s | 3.0591s | 3.7154s | 3.9062s | 4.4952s |
| Pose estimation | 20.6085s | 24.0013s | 27.4166s | 30.8672s | 35.3407s |
| RVC | 5.9726s | 6.4080s | 7.5483s | 10.3359s | 12.2985s |
| Overall | 33.9525s | 38.3310s | 43.9903s | 50.9244s | 58.6605s |

To demonstrate the efficiency of our method in detail, we report the inference time for each component on an NVIDIA L20 GPU (46GB memory) with different numbers of input images in Tab. 10. As shown in the table, the dominant runtime contributors are pose estimation and RVC. Nevertheless, the total inference time remains reasonable. The RVC is a justified trade-off for accuracy and efficiency. RVC provides a significant quality boost of approximately $1.587$ dB PSNR compared to the pipeline without it. This gain is crucial for achieving high-fidelity 3D reconstruction and eliminating visual inconsistencies in input images. Importantly, even without pose estimation and RVC, the results still surpass current SOTA reconstruction methods, while RVC further improves fine geometry and texture alignment.

## A.11 THEORETICAL EXPLANATION ON HOW RVC INTERACTS WITH RECTIFIED FLOW OBJECTIVES

RVC does not modify the underlying rectified-flow training objective or the learned vector field $v_\theta$. Instead, it is an inference-time correction mechanism that adjusts the denoising trajectory according

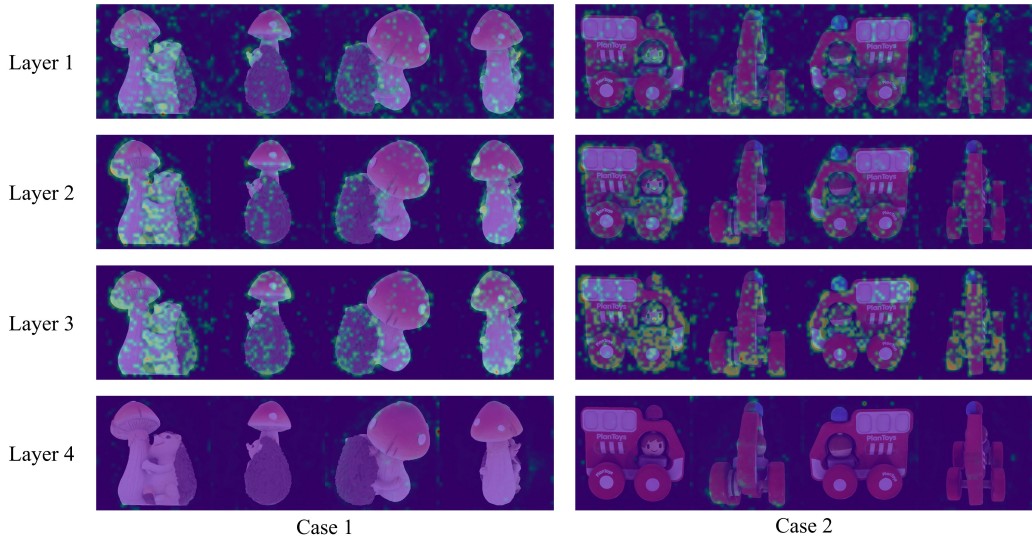

Figure 4: Attention visualization in GGC.

to rendering errors from the input views. Concretely, the rectified flow predicts the direction of the straightened transport path as:

$$x_{t-\Delta t} = x_t - \Delta t \, v_\theta(x_t, t). \tag{1}$$

RVC introduces a small correction:

$$x_{t-\Delta t} = x_t - \Delta t(v_\theta + \alpha \, \Delta v), \tag{2}$$

where $\Delta v = -t \frac{\partial L_{\text{RVC}}}{\partial x_0}$. This adjustment is orthogonal to the rectified-flow objective. The flow model is fully trained under the standard CFM loss. And RVC only guides the denoising trajectory toward a solution that is better aligned with input images. The correction does not alter the learned velocity field or its theoretical properties. In summary, RVC provides input-conditioned guidance during inference, but does not change the theoretical formulation or training of rectified flow.

## A.12 ABLATION STUDY ON DECODING TO DIFFERENT 3D REPRESENTATIONS IN RVC

Table 11: Ablation study on decoding to different 3D representations in RVC on the Dora-bench dataset.

| 3d representation | PSNR↑ | SSIM↑ | LPIPS↓ | CD↓ | F-score↑ |
|---|---|---|---|---|---|
| Radiance Field (RF) | 21.899 | 0.908 | 0.0910 | 0.0917 | 0.937 |
| Mesh | 21.561 | 0.906 | 0.0924 | **0.0879** | **0.954** |
| 3DGS | **22.632** | **0.911** | **0.0901** | 0.0895 | 0.953 |

To investigate the influence of 3D representation in RVC, we decode the output SLAT to different 3D representations and report their metrics on the Dora-Bench dataset. In Tab. 11, mesh shows slightly better geometric accuracy due to its structured nature. 3DGS provides superior rendering quality, leading to the best overall visual fidelity. Based on its superior rendering quality and overall balanced performance, choosing 3DGS as the representation is more appropriate.

## A.13 ATTENTION VISUALIZATIONS IN GGC AND CROSS-VIEW TOKEN SIMILARITY IN PVC

To fully understand the proposed GGC and PVC, we provide attention visualization in GGC and cross-view token similarity in PVC. Specifically, for attention visualization in GGC, we first average attention maps along the dimension of multi-head in each attention layer, and then max pool the attention maps along the dimension of learnable tokens. Finally, we resize the attention map as the same size as input images. For cross-view token similarity in PVC, we calculate the similarity

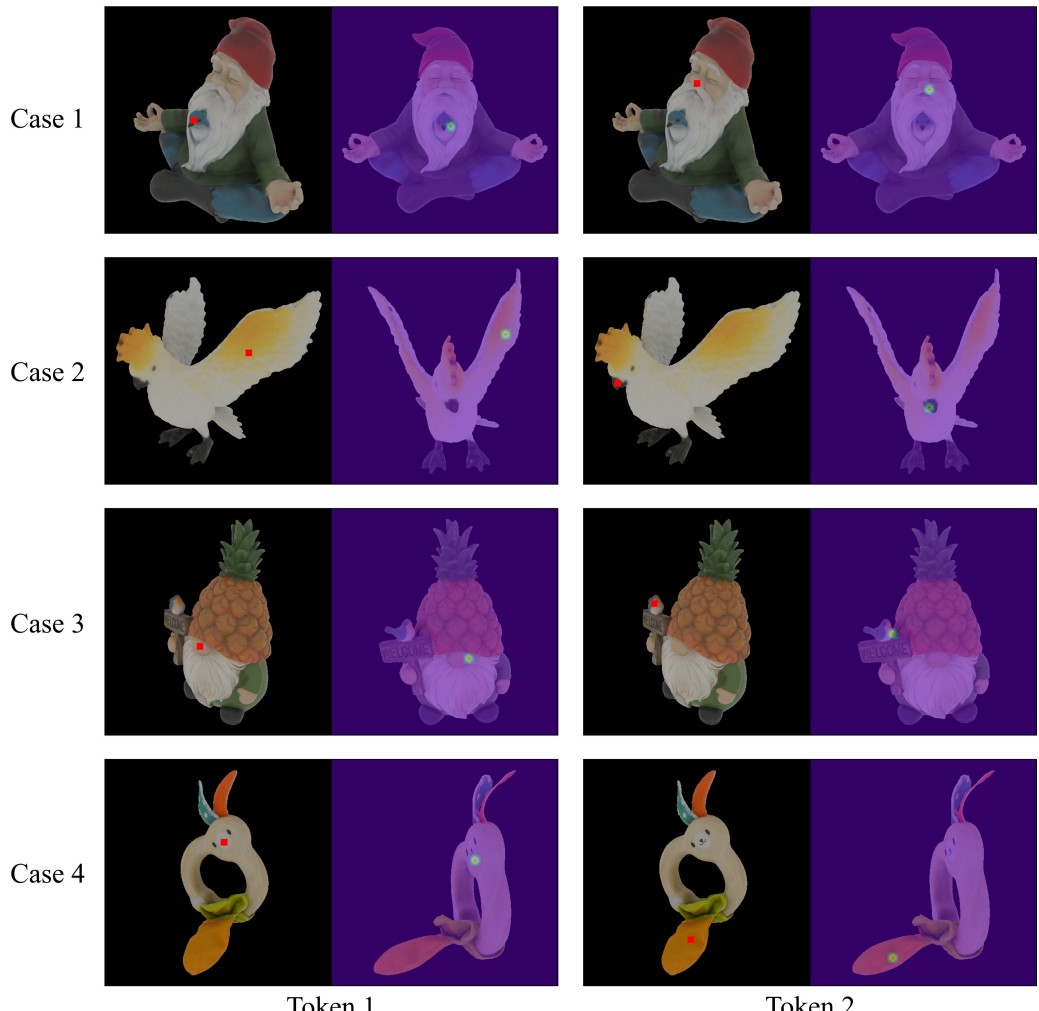

Case 1

Case 2

Case 3

Case 4

Token 1                                    Token 2

Figure 5: Cross-view token similarity in PVC.

between tokens of two different input images and select a token from one image to visualize its similarity to tokens of another image. As shown in Fig. 4, for GGC's global aggregation in SS, attention visualizations of the first three layers all show a strong correlation between the global learnable tokens and object features in images. In Fig. 5, for PVC's local correspondence in SLAT, we find that there is high similarity between corresponding points across different viewpoints.

A.14 ABLATION STUDY ON THE LENGTH OF LEARNABLE TOKENS IN GGC

Table 12: Ablation study on the length of learnable tokens in GGC on the Dora-bench dataset.

| Length of tokens | PSNR↑ | SSIM↑ | LPIPS↓ | CD↓ | F-score↑ |
|---|---|---|---|---|---|
| 2048 | 18.366 | 0.890 | 0.107 | 0.114 | 0.884 |
| 4096 | 20.462 | **0.894** | 0.102 | 0.093 | **0.941** |
| 8192 | **20.527** | 0.887 | **0.105** | **0.092** | 0.938 |

We conduct the ablation study on the length of learnable tokens in GGC. Since the length of SS latent is $16^3 = 4096$, we believe 4096 is enough for the length of learnable tokens in GGC to represent the visible structure. As shown in Tab. 12, the significant performance drop shows that 2048 tokens are insufficient to represent the information of input. Although 8192 tokens offer a

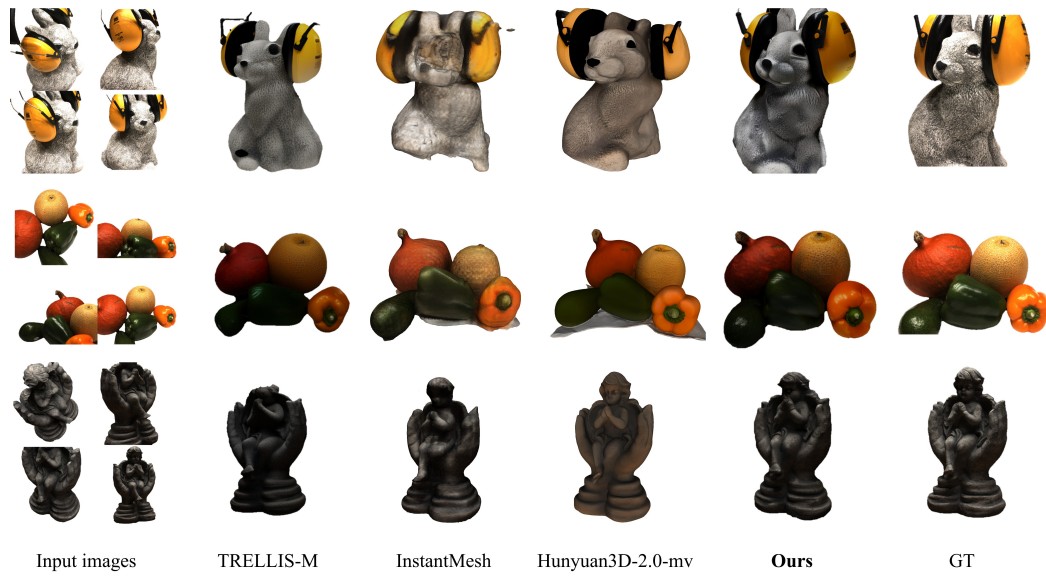

| Input images | TRELLIS-M | InstantMesh | Hunyuan3D-2.0-mv | **Ours** | GT |

Figure 6: Qualitative comparisons on the DTU datset.

slight improvement in PSNR and CD, they introduce additional computational overhead. Therefore, 4096 learnable tokens provide the balance between performance and efficiency.

## A.15 EVALUATION ON THE DTU DATASET

Table 13: Evaluation on the DTU dataset.

| Method | PSNR↑ | SSIM↑ | LPIPS↓ |
|---|---|---|---|
| TRELLIS-M Xiang et al. (2024) | 15.8675 | 0.6010 | 0.3237 |
| Hunyuan3D-2.0-mv Team (2025) | 19.0951 | 0.6763 | 0.2654 |
| Instantmesh Xu et al. (2024) | 18.8966 | 0.6712 | 0.2527 |
| Ours | **21.7639** | **0.7576** | **0.2175** |

DTU Jensen et al. (2014) is a real-world multi-view dataset. To further verify the generalizability of our method on real multi-view captures, we evaluate three SOTA multi-view reconstruction baselines (TRELLIS-M, InstantMesh, and Hunyuan3D-2.0-mv) and our method on the DTU dataset, and the results are summarized in Tab. 13. Our approach substantially improves reconstruction fidelity on real-world multi-view data (+2.7 PSNR over the strongest baseline). These results confirm that ReconViaGen generalizes well to real multi-view captures, without any dataset-specific tuning. Qualitative comparisons in Fig. 6 further demonstrate the superiority of our ReconViaGen in real multi-view captures.

## A.16 ABLATION STUDY ON RVC ONLY BASELINE WITH OUR FULL-VERSION MODEL

Table 14: Ablation on RVC only baseline with our full-version model.

| Method | PSNR↑ | SSIM↑ | LPIPS↓ | CD↓ | F-score↑ |
|---|---|---|---|---|---|
| TRELLIS-M Xiang et al. (2024) | 16.706 | 0.882 | 0.111 | 0.144 | 0.843 |
| TRELLIS-M Xiang et al. (2024) w. RVC | 17.728 | 0.857 | 0.101 | 0.139 | 0.851 |
| Ours | **22.632** | **0.911** | **0.090** | **0.090** | **0.953** |

We conduct experiments of TRELLIS-M with RVC on Dora-Bench. As shown in the Tab. 14, RVC alone consistently improves the baseline, demonstrating that RVC is indeed effective. However, its performance remains far below our full model, since RVC mainly provides local refinement and

cannot replace the global structure aggregation (GGC) and per-view detail modeling (PVC). This confirms that RVC is beneficial but complements rather than substitutes the other components.

## A.17 Ablation study on alternative design for weighted fusion in SLat-Flow

Table 15: Ablation on alternative design for weighted fusion in SLat-Flow on the Dora-bench dataset.

| Method | PSNR↑ | SSIM↑ | LPIPS↓ | CD↓ | F-score↑ |
|---|---|---|---|---|---|
| Add up + MLP | 20.809 | 0.894 | **0.091** | 0.098 | 0.924 |
| Our weighted fusion (c) | **21.045** | **0.905** | 0.093 | **0.093** | **0.937** |

The weighted fusion design aims to trade off multi-view specificity and computational efficiency. The SLAT latent already encodes global geometry and acts as a "volume memory". Therefore, the result of cross attention between SLAT latent and each individual view already reflects how much that view contributes to the current latent state. The MLP is then used to map this interaction to a scalar weight, enabling view-dependent importance estimation without requiring joint attention over all views. Another alternative design is to add all the cross-attention results from all views together and then apply an MLP. We ablation on this alternative design with ours in Tab. 15. The alternative yields worse performance. We think direct summation reduces view-specific differences, confirming the need for a view-aware weighting scheme. A direct way is to apply a cross-attention with the KV cache coming from all the views. We experimented with this design, but the KV cache grows linearly with the number of views and quickly exceeds GPU memory. Thus, per-view cross-attention with a learned weight provides the balance of accuracy, interpretability, and memory efficiency.

## A.18 Clarification for omitting the visual metrics of VGGT in the Dora-Bench evaluation

Using point clouds from VGGT directly for image rendering does indeed produce poor results. For example, due to the discrete representation of point clouds, when rendering an object from the front, rays may pass through the front point clouds and hit the back point clouds instead. This is due to deficiencies in point cloud representation; we believe that reporting visual metrics in this way is unfair to VGGT. To use the output of VGGT for image rendering, a direct way is to combine the VGGT with 3DGS optimization. We specifically compared ours with VGGT+3DGS in Sec.A.2. However, 3DGS optimization introduces many artifacts and floaters, which negatively impact geometric metrics like CD and F-score, which is unfair to VGGT. 3D reconstruction often focuses more on geometry, so we choose to report only VGGT's geometric metrics and omit its visual metrics.

## A.19 Sensitivity analysis of the quality to camera pose accuracy

Table 16: Ablation study on camera poses under different extent of rotational perturbation on Dora Bench.

| Method | PSNR↑ | SSIM↑ | LPIPS↓ | CD↓ | F-score↑ |
|---|---|---|---|---|---|
| Ours w/o RVC | 21.045 | 0.905 | 0.093 | 0.093 | 0.937 |
| Ours w. GT pose | **23.957** | **0.917** | 0.092 | **0.086** | 0.955 |
| Ours w. 3° perturbation | 23.606 | 0.914 | **0.089** | 0.088 | **0.957** |
| Ours w. 5° perturbation | 22.788 | 0.916 | 0.093 | 0.092 | 0.951 |
| Ours w. 10° perturbation | 21.221 | 0.909 | 0.094 | 0.095 | 0.950 |
| Ours w. 30° perturbation | 20.939 | 0.903 | 0.094 | 0.093 | 0.941 |
| Ours w. 50° perturbation | 20.906 | 0.903 | 0.097 | 0.093 | 0.938 |
| Ours | 22.632 | 0.911 | 0.090 | 0.090 | 0.953 |

To investigate the robustness and sensitivity of RVC to camera pose accuracy, we intentionally introduce severe rotational perturbation to the ground-truth camera pose in RVC. As shown in Tab. 16, RVC improves reconstruction significantly when accurate poses are available. With moderate errors ($\leq 10$), the degradation is minimal, showing high tolerance. Even with extreme perturbations of $30°$ and $50°$ (far beyond practical estimation errors), the performance remains comparable to the baseline "Ours w/o RVC", demonstrating that RVC does not destabilize or corrupt the denoising trajectory. The key to this robustness lies in our outlier mechanism within the rendering-based loss.

### A.20 THE USE OF LARGE LANGUAGE MODELS

We only utilize LLMs to refine the writing style and enhance the clarity of exposition. The LLMs are not involved in research ideation, experimental design or data analysis.