# OpenReview forum: "ReconViaGen: Towards Accurate Multi-view 3D Object Reconstruction via Generation"
_ICLR.cc/2026/Conference — ICLR 2026 Poster_

### Official Review · Reviewer_3uhU · 2025-10-16

**Soundness:** 2
**Presentation:** 3
**Contribution:** 3
**Rating:** 6
**Confidence:** 4

**Summary:**

1. **Originality-wise**: The paper introduces a novel framework that uses implicit, 3D-aware features from a reconstruction network (VGGT) to condition a diffusion generator (TRELLIS) , and a rendering-aware feedback loop (RVC), correcting the denoising trajectory during inference. It represents a new and sophisticated approach to solving the trade-off between reconstruction completeness and generative accuracy.

2. **Quality-wise**: The claims are substantiated with a robust and comprehensive evaluation on challenging benchmarks like Dora-Bench and OmniObject3D.

3. **Clarity-wise**: The manuscript is clearly written, with well-structured methodology, detailed explanations, and intuitive visualizations that enhance understanding.

**Strengths:**

1. Clear problem formulation and strong motivation: The paper accurately identifies the central conflict in modern 3D reconstruction between the "incompleteness of reconstruction" and the "inaccuracy of generation." The analysis of the root causes—such as insufficient cross-view correlation and poor controllability of the generation process—provides a solid foundation for the proposed method.

2. Innovative and well-Designed framework: ReconViaGen is more than a simple cascade of reconstruction and generation models. It achieves a deep fusion of the two priors by integrating the feature-level output of the reconstructor directly into the conditioning mechanism of the diffusion generator. The coarse-to-fine strategy, guided by separate global and local conditions, is a logical and effective way to leverage multi-level reconstruction cues.

**Weaknesses:**

This paper is good writing and well visualized, but still with some minor weakness:

1. Concern about computational cost: While highly effective, the RVC mechanism requires 3D decoding, rendering, loss computation, and gradient calculation at multiple steps of the diffusion inference process. This will undoubtedly add significant computational overhead. The paper does not provide a detailed analysis of this overhead, which could be a potential bottleneck for applications requiring fast inference. A discussion on the trade-off between speed and accuracy would be beneficial.

2. The effectiveness of the RVC mechanism is critically dependent on the accuracy of the estimated camera poses. In challenging sparse-view scenarios where pose estimation is inherently less reliable, any inaccuracies will directly corrupt the rendering-based loss, providing erroneous guidance to the diffusion model. This risks actively misleading the denoising trajectory rather than correcting it. The paper lacks a crucial robustness analysis to quantify the sensitivity of the final reconstruction quality to errors in camera pose estimation.

---

I have listed my concerns, and the score will be adjusted based on the author's response.

**Questions:**

Please refer to Weaknesses part.

---

> ### Author Response · Authors · 2025-11-26
> **Official Response by Authors**
>
> We appreciate the insightful comments and positive support with constructive feedback. We hope our responses address the reviewer's concerns.
>
> ## Computational cost
>
> Thanks for your suggestion. We report the inference time of each component with various numbers of input images in Section A.11 of Appendix:
>
> |Component \ #images|1|3|5|7|9|
> |:----:|:----:|:----:|:----:|:----:|:----:|
> |DINO feature & VGGT feature|0.2157s|0.4393s|0.8789s|1.3788s|2.0745s|
> |GGC|0.0169s|0.0175s|0.0184s|0.0199s|0.0224s|
> |SS Flow & SS Decoder|4.4193s|4.4039s|4.4105s|4.4141s|4.4267s|
> |PVC|0.0017s|0.0019s|0.0022s|0.0023s|0.0025s|
> |SLat Flow & SLat Decoder|2.7178s|3.0591s|3.7154s|3.9062s|4.4952s|
> |Pose estimation|20.6085s|24.0013s|27.4166s|30.8672s|35.3407s|
> |RVC|5.9726s|6.4080s|7.5483s|10.3359s|12.2985s|
> |Overall|33.9525s|38.3310s|43.9903s|50.9244s|58.6605s|
>
> As shown in the table, the dominant runtime contributors are pose estimation and RVC. Nevertheless, the total inference time remains reasonable for high-fidelity sparse-view reconstruction.
> We argue that the RVC's computational cost is a justified trade-off for accuracy and robustness. RVC provides a significant quality boost of approximately 1.5 dB PSNR compared to the pipeline without it. This gain is crucial for achieving high-fidelity 3D reconstruction and eliminating visual inconsistencies to input images.
> Importantly, even without pose estimation and RVC, the backbone reconstruction pipeline already surpasses existing SOTA methods, while RVC further improves fine geometry and texture alignment.
>
> ## Quantify the sensitivity of the final reconstruction quality to errors in camera pose estimation
>
> Thanks for your advice. To investigate the robustness and sensitivity of RVC to camera pose accuracy, we intentionally introduce severe rotational perturbation to the ground-truth camera pose in RVC:
>
> |Method|PSNR $\uparrow$|SSIM $\uparrow$|LPIPS $\downarrow$|CD $\downarrow$ |F-score $\uparrow$|
> |:----:|:----:|:----:|:----:|:----:|:----:|
> |Ours w/o RVC|21.045|0.905|0.093|0.093|0.937|
> |Ours w. GT pose|23.957|0.917|0.092|0.086|0.955|
> |Ours w. 3° perturbation|23.606|0.914|0.089|0.088|0.957|
> |Ours w. 5° perturbation|22.788|0.916|0.093|0.092|0.951|
> |Ours w. 10° perturbation|21.221|0.909|0.094|0.095|0.950|
> |Ours w. 30° perturbation|20.939|0.903|0.094|0.093|0.941|
> |Ours w. 50° perturbation|20.906|0.903|0.097|0.093|0.938|
> |Ours|22.632|0.911|0.090|0.090|0.953|
>
> RVC improves reconstruction significantly when accurate poses are available.
> With moderate errors (≤10°), the degradation is minimal, showing high tolerance.
> Even with extreme perturbations 30° and 50° (far beyond practical estimation errors), the performance remains comparable to the baseline "Ours w/o RVC", demonstrating that RVC does not destabilize or corrupt the denoising trajectory.
> The key to this robustness lies in our outlier filter mechanism within the rendering-based loss. We include this experiment in the revised manuscript (Appendix A.20).

---

> ### Comment · Reviewer_3uhU · 2025-11-26
>
> Thanks for your reply and revision.
> The reply and extended exps have addressed my concerns.
> I will maintain my score and tend to accept this work.

---

> > ### Author Response · Authors · 2025-11-27
> > **Thanks for Your Review**
> >
> > Thanks for the constructive comments and the time you dedicate to the paper! Thanks a lot again, and with sincerest best wishes

---

### Official Review · Reviewer_LmLx · 2025-10-29

**Soundness:** 3
**Presentation:** 3
**Contribution:** 2
**Rating:** 4
**Confidence:** 5

**Summary:**

This paper proposes to incorporate VGGT [2] features into Trellis [1] generation process via simple cross attention in order to improve the adherence of the Trellis Generators (SS and SLAT) to the provided input views.

This paper demonstrates an interesting insight towards how the 3D global information aggregated by VGGT is more meaningful to the Trellis denoisers compared to providing the raw unprocessed RGB images directly, however it falls short of shedding a scientific light onto why this is the case. Moreover, the important high-frequency details in their results actually come from a highly empirical inference-time iterative refining step (RVC) which almost doubles the generation time (at the least, details missing), and not from the featurewise cross connection of VGGT and Trellis. Thus this research raises concerns about the actual utility of the proposed mutli-step pipeline. More detailed concerns in the Weakness section.



[1] Xiang, J., Lv, Z., Xu, S., Deng, Y., Wang, R., Zhang, B., ... & Yang, J. (2025). Structured 3d latents for scalable and versatile 3d generation. In Proceedings of the Computer Vision and Pattern Recognition Conference (pp. 21469-21480).

[2] Wang, J., Chen, M., Karaev, N., Vedaldi, A., Rupprecht, C., & Novotny, D. (2025). Vggt: Visual geometry grounded transformer. In Proceedings of the Computer Vision and Pattern Recognition Conference (pp. 5294-5306).

**Strengths:**

1. The authors combine two very strong recent works VGGT and TRELLIS with a clean Cross-Attention module into an E2E pipeline which yields decent improvements in the results.

2. The paper is presented in a clear and lucid manner.

**Weaknesses:**

The paper however raises the following concerns:

**Major Concerns**:
1. The RVC step contributes the most amount of visual details to the generated results, and as such, this step lacks sufficient details and discussion in the paper. Most importantly, how does an RVC only baseline fare against the `(d)-full` version of the method. For the Rendering aware correction, something simple as the predicted point-clouds of VGGT or even sparse reconstructions from COLMAP are meaningful because in this version no training is needed. Moreover, it would be quite important for this paper to present details about how the hyperparameters of the final step are tuned, and whether per-asset generation tuning is needed of if some fixed values could be set in the pipeline and then generalised for inference. Lastly, how much generation latency does the RVC step add to the full pipeline?

2. As alluded to earlier in the summary, imho, a discussion about why the global features from VGGT help the Trellis generation so much is imperative for this paper. Of course we can say that the learned features aggregated by VGGT through the regressive prior capture much more 3D structural information than just the 2D multi-views, but then the question that rises is how come the generative 3D prior implicitly learned by Trellis is not enough? If it's possible to entirely use Trellis (through finetuning), then the solution becomes even more cleaner than the current one which integrates two different large-scale foundation models.

**Minor concerns**:
1. Are the two ConditionNets for the global condition as well as the local per-view condition tokens the same? This is not clarified properly in the draft as the Mathematical notation is the same for the two, and this is not clarified explicitly.

2. What is the intuition behind the weighted fusion of the per-view features cross-attention conditioning? Just for computational efficiency? Details are missing for this particular operation. How can the MLP predict from just the current' view's features which of them are important and which aren't when fusing multiple views? Why not just add all the features together and them apply a linear layer/MLP? Ideally, the simplest operation should be applying a cross-attention with the KV cache coming from all the views, but might cause OOM. Clarification from the authors would be appreciated. An ablation on this would make the paper more well rounded.

3. How is the SLAT-FLow transformer trained? Similar to the SS-FLow? Implementation Details paragraph lines 350-359 do not cover this explicitly.

4. What is the total time E2E time required for the 3D model generation starting from Multi-view inputs to receiving the final 3D model output? How much the RCV contribute to the generation latency? A detailed latency table would be useful for demonstrating the utility of the final proposed pipeline.

5. I think it would be very beneficial to include more qualitative examples for the (c) v/s (d) ablation study. It will be very insightful to understand where exactly the ~1.5 dB improvement is coming from.

**Questions:**

Responses to the Major Concerns 1 and 2 would be most informative to me in making the final decision in the discussion phase.

**Minor Suggestions**:
1. The visual metrics (PSNR, LPIPS and SSIM) for the VGGT baseline, however low, will be helpful for completeness of the scientific study. They have been possibly left out because of the low numbers coming from the holes in the renders of VGGT, but not detailed why exactly they have been left out. Clarification appreciated.

---

> ### Author Response · Authors · 2025-11-26
> **Official Response by Authors -- Part 1**
>
> We would like to sincerely thank the reviewer for the valuable and constructive comments on our work. We take every comment seriously and hope our response can address the reviewer’s concerns.
>
> ## RVC only baseline fare against the (d)-full version method
>
> Thanks for your advice. We conduct experiments of TRELLIS-M with RVC on Dora-Bench:
>
> |Method| PSNR $\uparrow$|SSIM $\uparrow$|LPIPS $\downarrow$|CD $\downarrow$ |F-score $\uparrow$|
> |:----:|:----:|:----:|:----:|:----:|:----:|
> |TRELLIS-M|16.706|0.882|0.111|0.144|0.843|
> |TRELLIS-M w. RVC|17.728|0.857|0.101|0.139|0.851|
> |Ours|22.632|0.911|0.090|0.090|0.953|
>
> As shown in the table, RVC alone consistently improves the baseline, demonstrating that RVC is indeed effective. However, its performance remains far below our full model, since RVC mainly provides local refinement and cannot replace the global structure aggregation (GGC) and per-view detail modeling (PVC).
> This confirms that RVC is beneficial but complements rather than substitutes the other components. We have added this clarification in the revised manuscript (Appendix A.17).
>
> ## Predicted point-clouds of VGGT or even sparse reconstructions from COLMAP for RVC
>
> Thank you for the suggestion. First, point cloud is difficult to be adopted for rendering aware correction. A reasonable approach is to use point cloud from VGGT as reference and apply the Dice loss to correct the denoising process of SS:
>
> |Method| PSNR $\uparrow$|SSIM $\uparrow$|LPIPS $\downarrow$|CD $\downarrow$ |F-score $\uparrow$|
> |:----:|:----:|:----:|:----:|:----:|:----:|
> |(b)|20.462|0.894|0.102|0.093|0.941|
> |(b) w. pc guidance|20.0191|0.869|0.1115|0.102|0.926|
>
> However, under the sparse view setting, these point clouds from VGGT are too inaccurate to serve as reliable supervision.
> As shown in our experiments, adding VGGT point-cloud guidance actually hurts reconstruction.
> COLMAP reconstruction is even less stable and frequently fail in sparse views.
> Therefore, we find that image-based correction is substantially more robust and effective than point-cloud–based guidance in sparse-view scenarios.
>
> ## Details about how the hyperparameters tuned and whether per-asset generation tuning is needed
>
> Thanks for advice. We provide the ablation study on the extent of RVC $\alpha$, timestep t, and outlier rejection threshold $t_o$ in Section A.10 of Appendix:
>
> |$\alpha$|PSNR $\uparrow$|SSIM $\uparrow$|LPIPS $\downarrow$|CD $\downarrow$ |F-score $\uparrow$|
> |:----:|:----:|:----:|:----:|:----:|:----:|
> |1|22.475|0.911|0.091|0.091|0.941|
> |0.5|22.151|0.913|0.093|0.089|0.949|
> |0.1|22.632|0.911|0.090|0.090|0.953|
> |0.05|21.321|0.910|0.089|0.091|0.953|
> |0.01|21.079|0.906|0.095|0.092|0.941|
>
> |t| PSNR $\uparrow$|SSIM $\uparrow$|LPIPS $\downarrow$|CD $\downarrow$ |F-score $\uparrow$|Time $\downarrow$ |
> |:----:|:----:|:----:|:----:|:----:|:----:|:----:|
> |0.7|22.149|0.907|0.090|0.094|0.938|9.2767|
> |0.6|22.477|0.912|0.091|0.094|0.940|8.2999|
> |0.5|22.632|0.911|0.090|0.090|0.953|6.8652|
> |0.4|22.270|0.909|0.089|0.089|0.944|5.7155|
> |0.3|22.057|0.915|0.088|0.091|0.937|4.7825|
>
> |$t_o$| PSNR $\uparrow$|SSIM $\uparrow$|LPIPS $\downarrow$|CD $\downarrow$ |F-score $\uparrow$|
> |:----:|:----:|:----:|:----:|:----:|:----:|
> |1|21.618|0.912|0.094|0.091|0.938|
> |0.9|22.075|0.906|0.090|0.091|0.946|
> |0.8|22.632|0.911|0.090|0.090|0.953|
> |0.7|22.450|0.909|0.094|0.089|0.941|
> |0.6|22.312|0.906|0.093|0.094|0.944|
>
>
> The results show that our model is robust across a wide range of settings, with only small performance variations. For $\alpha$, varying α from 1 to 0.01, we observe stable performance across a wide range, with the best results around α = 0.1. Extremely small values slightly reduce accuracy, but no instability was observed. Importantly, α introduces no runtime overhead, as it only changes the update weight of a vector. Applying RVC at different timesteps shows that t=0.5 offers the best trade-off between fidelity and cost. As expected, runtime scales linearly with the fraction of steps with RVC (due to decoding/rendering), ranging from 4.8s (t=0.3) to 9.3s (t=0.7). For outlier rejection threshold $t_o$, results are insensitive across a broad range. We choose $t_o=0.8$ as it yields strong PSNR/F-score. Therefore, α=0.1, t=0.5, and $t_o=0.8$ is the optimal combination of hyperparameters. Importantly, the best-performing configuration ($α = 0.1, t=0.5, t_o=0.8$) is shared across all assets and datasets, and no per-asset tuning is required.
> This confirms that the final RVC step generalizes well and does not rely on instance-specific adjustments.

---

> ### Author Response · Authors · 2025-11-26
> **Official Response by Authors -- Part 2**
>
> ## How much generation latency does the RVC step add to the full pipeline
>
> Thanks for your advice. We report the inference time of each component with various numbers of input images in Section A.11 of Appendix:
>
> |Component \ #images|1|3|5|7|9|
> |:----:|:----:|:----:|:----:|:----:|:----:|
> |DINO feature & VGGT feature|0.2157s|0.4393s|0.8789s|1.3788s|2.0745s|
> |GGC|0.0169s|0.0175s|0.0184s|0.0199s|0.0224s|
> |SS Flow & SS Decoder|4.4193s|4.4039s|4.4105s|4.4141s|4.4267s|
> |PVC|0.0017s|0.0019s|0.0022s|0.0023s|0.0025s|
> |SLat Flow & SLat Decoder|2.7178s|3.0591s|3.7154s|3.9062s|4.4952s|
> |Pose estimation|20.6085s|24.0013s|27.4166s|30.8672s|35.3407s|
> |RVC|5.9726s|6.4080s|7.5483s|10.3359s|12.2985s|
> |Overall|33.9525s|38.3310s|43.9903s|50.9244s|58.6605s|
>
> As shown in the table, the dominant runtime contributors are pose estimation and RVC.
> Nevertheless, the total inference time remains reasonable for high-fidelity sparse-view reconstruction.
> We argue that the RVC's computational cost is a justified trade-off for accuracy and efficiency. RVC provides a significant quality boost of approximately $1.5$ dB PSNR compared to the pipeline without it. This gain is crucial for achieving high-fidelity 3D reconstruction and eliminating visual inconsistencies to input images.
> Importantly, even without pose estimation and RVC, the backbone reconstruction pipeline already surpasses existing SOTA methods, while RVC further improves fine geometry and texture alignment.
>
> ## Why the global features from VGGT help the Trellis generation, 3D prior implicitly learned by Trellis is not enough, and possible to entirely use Trellis, then the solution becomes even more cleaner
>
> Thank you for this important question.
>
> By comparing the TRELLIS and ours using single-view input on Dora-Bench:
>
> |Method|PSNR $\uparrow$|SSIM $\uparrow$|LPIPS $\downarrow$|CD $\downarrow$ |F-score $\uparrow$|
> |:----:|:----:|:----:|:----:|:----:|:----:|
> |TRELLIS |15.264|0.858|0.182|0.162|0.781|
> |Ours|18.438|0.887|0.106|0.135|0.838|
>
> Our model can reconstruct substantially more accurate 3D geometry and appearance due to the strong 3D-aware global cues injected from VGGT.
> VGGT is trained with a regressive global 3D prior over billions of multi-view samples, enabling it to aggregate multi-view information into a 3D-structurally consistent representation. Trellis, in contrast, learns an implicit 3D prior purely from its denoising trajectory, which is significantly weaker under single-view conditions. Therefore, the global features from VGGT can help the Trellis generation so much, and currently, we think there is room for improvement in the 3D prior implicitly learned by TRELLIS.
>
> In principle, we think it's possible to entirely finetune Trellis to learn sufficient 3D prior.
> While this requires sufficient interaction between images, similar to VGGT, which demands an excessively large number of parameters to be trained, large-scale multi-view training data, and significantly more computing resources.
> Our preliminary attempts at end-to-end finetuning were unstable and did not converge, likely due to data/compute limitations.
> Since VGGT already provides a well-optimized multi-view 3D prior, directly leveraging it is much more stable and efficient.
> We agree that a unified end-to-end framework is an appealing future direction and plan to explore this further.
>
> ## Whether the two ConditionNets are the same
>
> We are sorry for the confusion. The two ConditionNets are not the same and differ functionally.
> In GGC, the ConditionNet uses cross-attention between learnable tokens and multi-view features so that the learnable tokens can progressively integrate VGGT’s multi-level information.
> In PVC, the ConditionNet operates in a per-view manner, using cross-attention to fuse the multi-level VGGT features of each view into view-dependent tokens, which preserves high-frequency details.
> To avoid confusion, we change the mathematical notation of the two ConditionNets into two different symbols in the revised manuscript (main text 3.2 Reconstruction-based conditioning).

---

> ### Author Response · Authors · 2025-11-26
> **Official Response by Authors -- Part 3**
>
> ## Intuition behind the weighted fusion and add all the features together followed by a linear layer/MLP
>
> Thanks for your advice.
> The weighted fusion design aims to trade off multi-view specificity and computational efficiency.
> The SLAT latent already encodes global geometry and acts as a “volume memory”. Therefore, the result of cross attention between SLAT latent and each individual view already reflects how much that view contributes to the current latent state.
> The MLP is then used to map this interaction to a scalar weight, enabling view-dependent importance estimation without requiring joint attention over all views.
>
> We test this alternative (“add up & MLP”):
>
> |Method|PSNR $\uparrow$|SSIM $\uparrow$|LPIPS $\downarrow$|CD $\downarrow$ |F-score $\uparrow$|
> |:----:|:----:|:----:|:----:|:----:|:----:|
> |add up & MLP|20.809|0.894|0.091|0.098|0.924|
> |our weighted fusion (c)|21.045|0.905|0.093|0.093|0.937|
>
> The alternative yields worse performance.
> We think direct summation reduces view-specific differences, confirming the need for a view-aware weighting scheme.
> For applying a cross-attention with the KV cache coming from all the views, we experimented with this design, but the KV cache grows linearly with the number of views and quickly exceeds GPU memory, matching the reviewer’s concern.
> Thus, per-view cross-attention with a learned weight provides the balance of accuracy, interpretability, and memory efficiency.
> We include this experiment and discussion in the revised manuscript (Appendix A.18).
>
> ## Implementation details about traning of the SLAT-FLow transformer
>
> Thanks for your advice. The way of training SLAT-FLow transformer is broadly similar to the SS-FLow. The difference is that we use a slightly smaller batch size of 128 for SLAT-Flow, compared to 192 for SS-Flow. We add this training difference in the Implementation Details.
>
> ## A detailed latency table from inputs to the final 3D model
>
> Thanks for your suggestion. We report the inference time of each component with various numbers of input images in Section A.11 of Appendix:
>
> |Component \ #images|1|3|5|7|9|
> |:----:|:----:|:----:|:----:|:----:|:----:|
> |DINO feature & VGGT feature|0.2157s|0.4393s|0.8789s|1.3788s|2.0745s|
> |GGC|0.0169s|0.0175s|0.0184s|0.0199s|0.0224s|
> |SS Flow & SS Decoder|4.4193s|4.4039s|4.4105s|4.4141s|4.4267s|
> |PVC|0.0017s|0.0019s|0.0022s|0.0023s|0.0025s|
> |SLat Flow & SLat Decoder|2.7178s|3.0591s|3.7154s|3.9062s|4.4952s|
> |Pose estimation|20.6085s|24.0013s|27.4166s|30.8672s|35.3407s|
> |RVC|5.9726s|6.4080s|7.5483s|10.3359s|12.2985s|
> |Overall|33.9525s|38.3310s|43.9903s|50.9244s|58.6605s|
>
> As shown in the table, the dominant runtime contributors are pose estimation and RVC.
> Nevertheless, the total inference time remains reasonable for high-fidelity sparse-view reconstruction.
> We argue that the RVC is a justified trade-off for accuracy and efficiency. RVC provides a significant quality boost of approximately $1.587$ dB PSNR compared to the pipeline without it. This gain is crucial for achieving high-fidelity 3D reconstruction and eliminating visual inconsistencies to input images.
> Importantly, even without pose estimation and RVC, the backbone reconstruction pipeline already surpasses existing SOTA methods, while RVC further improves fine geometry and texture alignment.
>
> ## More qualitative examples for the (c) v/s (d) ablation study.
>
> Thanks for your advice. We supplement more qualitative examples and label the PSNR of each case for the (c) v/s (d) ablation study in the revised manuscript (Appendix A.9). The qualitative analysis confirms that the $\sim 1.5$ dB improvement is primarily due to RVC's ability to correct color and texture drift and enforcing high-frequency aligned with the input views.
>
> ## The visual metrics for the VGGT baseline
>
> Thanks for pointing it out.
> Using point clouds from VGGT directly for image rendering does indeed produce poor results. For example, due to the discrete representation of point clouds, when rendering an object from the front, rays may pass through the front point clouds and hit the back point clouds instead.
> This is due to deficiencies in point cloud representation, therefore we believe that reporting visual metrics in this way is unfair to VGGT.
> To use the output of VGGT for image rendering, a direct way is to combine the VGGT with 3DGS optimization.
> We specifically compared ours with VGGT+3DGS in Appendix A.2. However, 3DGS optimization introduces many artifacts and floaters, which negatively impacts geometric metrics like CD and F-score, which is unfair to VGGT. 3D reconstruction often focuses more on geometry, so we choose to report only VGGT's geometric metrics and omit its visual metrics. We add this clarification in the revised manuscript (Appendix A.19).

---

> > ### Comment · Reviewer_LmLx · 2025-11-26
> >
> > Thanks dear authors for your efforts and providing the detailed clarifications for the concerns I raised. I am happy to upgrade my rating for this paper.
> >
> > Details:
> > 1. Major concern about the RVC step in the full-generation pipeline, was clarified through ablation experiments on applying directly RVC on prior state demonstrating the utility of the full pipeline.
> > 2. Major concern about the need of a VGGT like prior also clarified through ablation experiment.
> >
> > I also appreciate the revision made to the paper towards my minor concerns, I believe it makes the paper more complete and well-rounded.
> >
> > Although, this approach is not ready for mass product deployment, I believe this will inspire the future researchers to think better in terms of connecting transformer based pipelines through clean modules, into achieving robust and useful applications.

---

> > > ### Author Response · Authors · 2025-11-27
> > > **Thanks for Your Review**
> > >
> > > Thank you for your response! We are delighted to hear that our reply addressed your concerns and appreciate your recognition of our work.

---

### Official Review · Reviewer_eFoP · 2025-10-31

**Soundness:** 3
**Presentation:** 3
**Contribution:** 3
**Rating:** 6
**Confidence:** 4

**Summary:**

The paper proposes ReconViaGen, a coarse-to-fine framework that integrates pose-free multi-view reconstruction priors (VGGT) with diffusion-based 3D generative priors (TRELLIS) for accurate and complete 3D object reconstruction from sparse, uncalibrated views. The method: (1) builds multi-view-aware conditions from VGGT via a ConditionNet, forming a Global Geometry Condition (GGC) for coarse Structure (SS) Flow and Per-View Conditions (PVC) for fine SLAT Flow; (2) introduces a rendering-aware velocity compensation (RVC) that, during inference, adjusts rectified-flow velocities using SSIM/LPIPS/DreamSim losses between rendered views and inputs; (3) refines camera poses by registering inputs into the generator’s space through rendering, VGGT estimation, feature matching, and PnP-RANSAC. Experiments on Dora-Bench and OmniObject3D show consistent SOTA improvements on image consistency (PSNR/SSIM/LPIPS), geometry (CD), and completeness (F-score), with ablations isolating the benefits of GGC/PVC/RVC and robustness to number/quality of views.

**Strengths:**

1.	Injects reconstruction priors (VGGT) into a 3D diffusion generator, addressing a well-known failure mode of existing generative methods—global structure drift & local inconsistency.
2.	The clear division of labor and strong motivation: GGC for structural cues and PVC for fine details match information granularity.
3.	RVC corrects the diffusion velocity field with differentiable rendering feedback at inference, simple to implement yet clearly improves input consistency.
4.	Broad evaluation with both image-consistency and geometry metrics plus pose accuracy; ablations are comprehensive and convincing.

**Weaknesses:**

1.	While results support “GGC is more suitable for SS and PVC is more suitable for SLAT,” deeper interpretability is missing.
2.	RVC hyperparameter/stability analysis is light: sensitivity to α, timestep threshold (t < 0.5), and the outlier rejection threshold (0.8) is not quantified; inference-time overhead (decoding/rendering per step) is not reported.
3.	Pose refinement pipeline is engineering-heavy.
4.	RVC is inference-only, creating a potential train–inference gap.
5.	Some theoretical clarity on how RVC interacts with rectified flow objectives is missing; decoding to different 3D representations (RF/3DGS/Mesh) and their effect on metrics could be further analyzed.

**Questions:**

1.	Provide attention visualizations and cross-view token similarity to show GGC’s global aggregation in SS and PVC’s local correspondence in SLAT?
2.	Conduct “Information bottleneck” experiments in SS (vary GGC token capacity) to study structure convergence and downstream SLAT performance?
3.	Provide sensitivity studies for RVC: α, timestep scheduling (t threshold), loss weights, and the 0.8 outlier discard rule? How do these affect convergence, runtime, and stability?
4.	Whether RVC, being inference-only, creates a potential train–inference gap. Why not introduce a lightweight RVC-like self-distillation during training to reduce distribution shift?
5.	Report performance on real multi-view captures (e.g., DTU object subset or an in-house dataset) to validate real-world generalization?

---

> ### Author Response · Authors · 2025-11-26
> **Official Response by Authors -- Part 1**
>
> We appreciate the insightful comments and positive support with constructive feedback. We hope our responses address the reviewer's concerns.
>
> ## Interpretability of GGC and PVC is missing
>
> Thank you for the suggestion. We clarify the interpretability behind why GGC benefits SS while PVC benefits SLAT. GGC performs global multi-view aggregation, which strengthens overall structural consistency—this is exactly what SS needs, since SS focuses on coarse geometry where global cues dominate and multi-view compression is acceptable. In contrast, SLAT requires fine-grained appearance and local geometry refinement on a fixed coarse shape. PVC’s per-view feature interaction preserves high-frequency cues that GGC would smooth out. Therefore, GGC naturally aligns with global structure formation, while PVC is better suited for detail-aware refinement. We add this explanation to the revised manuscript (Appendix A.5).
>
>
> ## RVC hyperparameter analysis
>
> Thanks for your advice. We provide the ablation study on the extent of RVC $\alpha$, timestep t, and outlier rejection threshold $t_o$ in Section A.10 of Appendix:
>
> |$\alpha$|PSNR $\uparrow$|SSIM $\uparrow$|LPIPS $\downarrow$|CD $\downarrow$ |F-score $\uparrow$|
> |:----:|:----:|:----:|:----:|:----:|:----:|
> |1|22.475|0.911|0.091|0.091|0.941|
> |0.5|22.151|0.913|0.093|0.089|0.949|
> |0.1|22.632|0.911|0.090|0.090|0.953|
> |0.05|21.321|0.910|0.089|0.091|0.953|
> |0.01|21.079|0.906|0.095|0.092|0.941|
>
> |t| PSNR $\uparrow$|SSIM $\uparrow$|LPIPS $\downarrow$|CD $\downarrow$ |F-score $\uparrow$|Time $\downarrow$ |
> |:----:|:----:|:----:|:----:|:----:|:----:|:----:|
> |0.7|22.149|0.907|0.090|0.094|0.938|9.2767|
> |0.6|22.477|0.912|0.091|0.094|0.940|8.2999|
> |0.5|22.632|0.911|0.090|0.090|0.953|6.8652|
> |0.4|22.270|0.909|0.089|0.089|0.944|5.7155|
> |0.3|22.057|0.915|0.088|0.091|0.937|4.7825|
>
> |$t_o$| PSNR $\uparrow$|SSIM $\uparrow$|LPIPS $\downarrow$|CD $\downarrow$ |F-score $\uparrow$|
> |:----:|:----:|:----:|:----:|:----:|:----:|
> |1|21.618|0.912|0.094|0.091|0.938|
> |0.9|22.075|0.906|0.090|0.091|0.946|
> |0.8|22.632|0.911|0.090|0.090|0.953|
> |0.7|22.450|0.909|0.094|0.089|0.941|
> |0.6|22.312|0.906|0.093|0.094|0.944|
>
> For $\alpha$, varying α from 1 to 0.01, we observe stable performance across a wide range, with the best results around α = 0.1. Extremely small values slightly reduce accuracy, but no instability was observed. Importantly, α introduces no runtime overhead, as it only changes the update weight of a vector. Applying RVC at different timesteps shows that t=0.5 offers the best trade-off between fidelity and cost. As expected, runtime scales linearly with the fraction of steps with RVC (due to decoding/rendering), ranging from 4.8s (t=0.3) to 9.3s (t=0.7). For outlier rejection threshold $t_o$, results are insensitive across a broad range. We choose $t_o=0.8$ as it yields strong PSNR/F-score. Therefore, α=0.1, t=0.5, and $t_o=0.8$ is the optimal combination of hyperparameters.
>
> ## Inference-time overhead.
>
> We also report the inference time of each component in Section A.11 of Appendix:
>
> |Component \ #images|1|3|5|7|9|
> |:----:|:----:|:----:|:----:|:----:|:----:|
> |DINO feature & VGGT feature|0.2157s|0.4393s|0.8789s|1.3788s|2.0745s|
> |GGC|0.0169s|0.0175s|0.0184s|0.0199s|0.0224s|
> |SS Flow & SS Decoder|4.4193s|4.4039s|4.4105s|4.4141s|4.4267s|
> |PVC|0.0017s|0.0019s|0.0022s|0.0023s|0.0025s|
> |SLat Flow & SLat Decoder|2.7178s|3.0591s|3.7154s|3.9062s|4.4952s|
> |Pose estimation|20.6085s|24.0013s|27.4166s|30.8672s|35.3407s|
> |RVC|5.9726s|6.4080s|7.5483s|10.3359s|12.2985s|
> |Overall|33.9525s|38.3310s|43.9903s|50.9244s|58.6605s|
>
> As shown in the table, the dominant runtime contributors are pose estimation and RVC.
> Nevertheless, the total inference time remains reasonable.
> Even without pose estimation and RVC, the results still surpass current SOTA reconstruction methods.
>
> ## Pose refinement is engineering-heavy.
>
> To be honest, we initially attempted end-to-end pose estimation, but these attempts failed to achieve satisfactory results. Due to the need to achieve accurate camera poses for subsequent RVC, we adopt current scheme. We fully agree that streamlining the pipeline is an important future direction. We are currently exploring ways to integrate pose estimation into an end-to-end framework.

---

> ### Author Response · Authors · 2025-11-26
> **Official Response by Authors -- Part 2**
>
> ## Potential train-inference gap in RVC and why not self-distilation during training.
>
> Thanks for pointing it out. Actually, our RVC is inspired by generative models [1,2,3] which use optimization techniques to correct output latent during the denoising process. In our experiments, we find that the potential gap does not have a significant impact. In our view, in the training phase, we directly use the 3D ground-turth latent to supervise the output of SLat Flow, and in the inference phase, the RVC leverages loss terms based on the 2D input images. Crucially, these 2D input images are the projections of the decoded 3D ground-truth SLat. Therefore, the 2D image constraints are geometrically highly coupled and consistent with the 3D ground truth used during training. In essence, RVC is a self-supervised refinement process that utilizes the input images as a form of partial, geometrically consistent ground truth for fine-tuning the latent representation.
>
> For lightweight RVC-like self-distillation during training, just like Hi3DGen [4], we have made such an attempt to add self-distillation during training. This approach provides a slight performance improvement (PSNR $\uparrow$ :21.045 -> 21.214, CD $\downarrow$ :0.093 -> 0.092), but the training cost is too high (additional SLAT decoding during training easily causes out of memory). And RVC is less effective for training than per-case optimization in inference (PSNR $\uparrow$ :21.214 vs 22.632, CD $\downarrow$ :0.092 vs 0.089), so we ultimately chose to adopt RVC in inference.
>
> [1] Mokady, R., Hertz, A., Aberman, K., Pritch, Y., & Cohen-Or, D. (2023). Null-text inversion for editing real images using guided diffusion models. In Proceedings of the IEEE/CVF conference on computer vision and pattern recognition (pp. 6038-6047).
>
> [2] Chen, J., Zhang, Y., Zou, Z., Chen, K., & Shi, Z. (2025). Zero-shot image harmonization with generative model prior. IEEE Transactions on Multimedia.
>
> [3] Chen, J., Zhang, B., Tang, X., & Wonka, P. (2025). V2M4: 4D Mesh Animation Reconstruction from a Single Monocular Video. arXiv preprint arXiv:2503.09631.
>
> [4] Ye, C., Wu, Y., Lu, Z., Chang, J., Guo, X., Zhou, J., ... & Han, X. (2025). Hi3dgen: High-fidelity 3d geometry generation from images via normal bridging. arXiv preprint arXiv:2503.22236, 3, 2.
>
> ## Theoretical clarity on how RVC interacts with rectified flow objectives
>
> Thank you for pointing this out. RVC does not modify the underlying rectified-flow training objective or the learned vector field $v_\theta$. Instead, it is an inference-time correction mechanism that adjusts the denoising trajectory according to rendering errors from the input views. Concretely, rectified flow predicts the direction of the straightened transport path as:
>
> $x_{t-\Delta t} = x_t - \Delta t\,v_\theta(x_t,t).$
>
> RVC introduces a small correction:
>
> $x_{t-\Delta t} = x_t - \Delta t (v_\theta + \alpha\, \Delta v),$
>
> where $\Delta v = -t\, \frac{\partial L_{\mathrm{RVC}}}{\partial x_0}.$
>
> This adjustment is orthogonal to the rectified-flow objective. The flow model is fully trained under the standard CFM loss. And RVC only guides the denoising trajectory toward a solution that is better aligned with input images. The correction does not alter the learned velocity field or its theoretical properties. In summary, RVC provides input-conditioned guidance during inference, but does not change the theoretical formulation or training of rectified flow. We add this explanation in the revised manuscript (Appendix A.12).
>
> ## Decoding to different 3D representations and their metrics
>
> Thanks for your advice. In RVC, we decode to different 3D representations and report their metrics on Dora-Bench:
>
> |3D representation| PSNR $\uparrow$|SSIM $\uparrow$|LPIPS $\downarrow$|CD $\downarrow$ |F-score $\uparrow$|
> |:----:|:----:|:----:|:----:|:----:|:----:|
> |RF  |21.899|0.908|0.0910|0.0917|0.937|
> |Mesh|21.561|0.906|0.0924|0.0879|0.954|
> |3DGS|22.632|0.911|0.0901|0.0895|0.953|
>
> Mesh shows slightly better geometric accuracy due to its structured nature. 3DGS provides superior rendering quality, leading to the best overall visual fidelity. Based on its superior rendering quality and overall balanced performance, choosing 3DGS as the representation is more appropriate. We include this ablation study in the revised manuscript (Appendix A.13).

---

> ### Author Response · Authors · 2025-11-26
> **Official Response by Authors -- Part 3**
>
> ## Attention visualizations in SS and cross-view token similarity in SLAT
>
> Thanks for your advice. We have included detailed attention visualizations and cross-view token similarity in Section A.14 of the Appendix. Specifically, for attention visualization in GGC, we first average attention maps along the dimension of multi-head in each attention layer, and then max pool the attention maps along the dimension of learnable tokens. Finally, we resize the attention map as the same size as input images. For cross-view token similarity in PVC, we calculate the similarity between tokens of two different input images and select a token from one image to visualize its similarity to tokens of another image. For GGC's global aggregation in SS, attention visualizations of the first three layers all show a strong correlation between the global learnable tokens and object features in images. For PVC's local correspondence in SLAT, we found there is high similarity between corresponding points across different viewpoints.
>
> ## Conduct "Information bottleneck" experiments in SS
>
> Thanks for pointing it out. We supplement the ablation study on the length of learnable tokens in GGC:
>
> |Length of tokens| PSNR $\uparrow$|SSIM $\uparrow$|LPIPS $\downarrow$|CD $\downarrow$ |F-score $\uparrow$|
> |:----:|:----:|:----:|:----:|:----:|:----:|
> |2048|18.366|0.890|0.107|0.114|0.884|
> |4096|20.462|0.894|0.102|0.093|0.941|
> |8192|20.527|0.887|0.105|0.092|0.938|
>
> Since the length of SS latent is $16^3=4096$, we believe 4096 is enough for the length of learnable tokens in GGC to represent the visible structure. The significant performance drop shows that 2048 tokens are insufficient to represent information of input. Although 8192 tokens offer a slight improvement in PSNR and CD, they introduce additional computational overhead. Therefore, 4096 learnable tokens provide the balance between performance and efficiency. We include this experiment in the revised manuscript (Appendix A.15).
>
> ## Performance on real multi-view captures
>
> Thanks for your suggestion. We have added evaluations on real multi-view captures using the DTU dataset. We compare against three SOTA multi-view reconstruction baselines (TRELLIS-M, InstantMesh, and Hunyuan3D-2.0-mv). As shown below, our method achieves the best performance across all metrics:
>
> |Method| PSNR $\uparrow$|SSIM $\uparrow$|LPIPS $\downarrow$|
> |:----:|:----:|:----:|:----:|
> |TRELLIS-M|15.8675|0.6010|0.3237|
> |Hunyuan3D-2.0-mv|19.0951|0.6763|0.2654|
> |InstantMesh|18.8966|0.6712|0.2527|
> |Ours|21.7639|0.7576|0.2175|
>
> Our approach substantially improves reconstruction fidelity on real-world multi-view data (+2.7 PSNR over the strongest baseline).
> These results confirm that ReconViaGen generalizes well to real multi-view captures, without any dataset-specific tuning.
> Visual comparisons and more details are included in Appendix A.16.

---

> > ### Comment · Reviewer_eFoP · 2025-11-28
> > **Comment**
> >
> > Thanks for the detailed responses that address my concerns.  I maintain my original score and am inclined to accept this work.

---

> > > ### Author Response · Authors · 2025-12-01
> > > **Thanks for Your Review**
> > >
> > > Thank you for your response! We are delighted to hear that our reply addressed your concerns and appreciate your recognition of our work.

---

### Official Review · Reviewer_Zbjq · 2025-10-31

**Soundness:** 3
**Presentation:** 2
**Contribution:** 3
**Rating:** 6
**Confidence:** 4

**Summary:**

This paper proposes the ReconViaGen framework, which aims to combine strong reconstruction priors (from VGGT) with diffusion-based generative priors (from TRELLIS) to address the limitations of pure reconstruction methods that lead to incomplete results and pure generative methods that, while complete, lack consistency with the input view. The method employs a coarse-to-fine strategy: using Global Geometric Conditions (GGC) to guide coarse structure generation and Per-View Conditions (PVC) to guide fine detail generation. Additionally, a Rendering-aware Velocity Compensation (RVC) mechanism is introduced, which is used only during inference to ensure pixel-level alignment of the generated results with the input view. Experiments on Dora-Bench and OmniObject3D datasets demonstrate that this method achieves SOTA levels in both completeness and consistency.

**Strengths:**

- The paper provides clear and detailed descriptions of the methodology and experimental design, making it easy to understand and replicate.

- The proposed ReconViaGen framework effectively addresses the limitations of pure reconstruction and pure generative methods by combining their strengths. The coarse-to-fine strategy, utilizing Global Geometric Conditions (GGC) for coarse structure and Per-View Conditions (PVC) for fine details, allows for a more accurate and consistent 3D reconstruction from single images. The results on Dora-Bench and OmniObject3D datasets demonstrate significant improvements in both completeness and consistency compared to state-of-the-art methods.

**Weaknesses:**

- The requirements for multi-view inputs may limit the applicability of the method in scenarios where only single-view inputs are available. In practical applications, users often can only provide a single view. It is unclear how the method would perform with single-view inputs, and the paper does not discuss or analyze this aspect.
- In Section 4.3, the ablation study does not provide visual comparisons to illustrate the effects of different modules, relying solely on numerical results to demonstrate their effectiveness. This makes it difficult for readers to intuitively understand the impact of each module on the final results.

**Questions:**

- The paper uses multi-view inputs as conditions for generation, but the results do not seem significantly better than current single-view 3D model generation methods (like Direct3D-S2[1]). Have the authors tried using single-view inputs as conditions for generation? If so, what were the results? I think it would be interesting to see how the model performs with single-view inputs compared to multi-view inputs.
- When using VGGT as the feature extractor for multi-view inputs, if the generated multi-view images are inconsistent, how much does this affect the quality of the final 3D model?

[1] Direct3D‑S2: Gigascale 3D Generation Made Easy with Spatial Sparse Attention

---

> ### Author Response · Authors · 2025-11-26
> **Official Response by Authors**
>
> We appreciate the insightful comments and positive support with constructive feedback. We hope our responses address the reviewer's concerns.
>
> ## Performance with single-view input.
>
> Thanks for your advice. Our primary research objective is to achieve accurate 3D object reconstruction, which aims at digitizing real-world objects with high fidelity. Single-view reconstruction is an inherently ill-posed problem, where the missing geometry and depth information will be hallucinated by the model. This ambiguity limits the achievable accuracy and primarily yields plausible results. Hence, we leverage the multi-view images to overcome the ill-posed nature of the problem, leading to the superior geometric consistency and fidelity shown in the paper.
> Despite our primary focus, our method is designed with a strong prior from VGGT which makes it highly effective even when constrained to a single input view. Due to time constraints in the rebuttal, we select two representative single-view generation baselines (TRELLIS and Direct3D-S2) for comparison on Dora-Bnech.
> As shown below, Ours substantially outperforms TRELLIS and Direct3D-S2:
> |Method|PSNR $\uparrow$|SSIM $\uparrow$|LPIPS $\downarrow$|CD $\downarrow$ |F-score $\uparrow$|
> |:----:|:----:|:----:|:----:|:----:|:----:|
> |TRELLIS |15.264|0.858|0.182|0.162|0.781|
> |Direct3D-S2|-|-|-|0.165|0.805|
> |Ours|18.438|0.887|0.106|0.135|0.838|
>
> This superior performance in the single-view setting highlights the versatility and the robustness of our method, confirming that ours is highly effective even with single-view input. More details and qualitative comparisons to demonstrate the superiority of our ReconViaGen on Dora-Bench and in-the-wild case are shown in the revised manuscript (Section A.8 of Appendix).
>
> ## No visual comparison of ablation study.
>
> The visual comparisons of ablation study are shown in Figure 5 of the main text. By comparing (a) with (b), we find that GGC significantly improves the global geometric structure, correcting coarse shape artifacts. By comparing (b) with (c), PVC mainly enhances appearance fidelity, yielding sharper textures and better view-consistency. By comparing (c) with (d), RVC further refines both geometry and texture alignment with the input views. We include more visualization of ablation study in the revised manuscript (Appendix A.9).
>
> ## Single-view input compared to multi-view inputs
>
> We provide a systematic study by varying the number of input views (1, 2, 4, 6, and 8). The results are shown below:
>
> |# Images|PSNR $\uparrow$|SSIM $\uparrow$|LPIPS $\downarrow$|CD $\downarrow$ |F-score $\uparrow$|
> |:----:|:----:|:----:|:----:|:----:|:----:|
> |1|18.438|0.887|0.106|0.135|0.838|
> |2|19.568|0.894|0.099|0.131|0.867|
> |4|22.632|0.911|0.090|0.090|0.953|
> |6|22.823|0.912|0.089|0.084|0.958|
> |8|23.067|0.914|0.090|0.081|0.961|
>
> More input views consistently improve performance, as expected—geometry becomes more complete and textures become more view-consistent. Additional details and visualizations are shown in Appendix A.4.
>
> ## Inconsistency of generated multi view images.
>
> Thank you for the question. Our primary research focuses on multi-view reconstruction, therefore our input is typically real-world multi-view images. To directly assess the effect of inconsistent multi-view inputs, we conducted an additional experiment using 6-view images generated by the multi-view generator Hunyuan3D-1.0. We combine these results with the single-view and real 6-view settings from Dora-Bench:
>
> |Method|PSNR $\uparrow$|SSIM $\uparrow$|LPIPS $\downarrow$|CD $\downarrow$ |F-score $\uparrow$|
> |:----:|:----:|:----:|:----:|:----:|:----:|
> |ours (generated 6-view)|14.379|0.808|0.226|0.190|0.723|
> |ours (single-view)|18.438|0.887|0.106|0.135|0.838|
> |ours (real 6-view) |22.823|0.912|0.089|0.084|0.958|
>
> In our observations, the results of generated 6 views as input are much worse than those of single-view and real 6-view input due to much severe inconsistency in multi-view generation.
> However, when the inconsistency in generated multi-view images is not severe, visualizations in Figure 7 show that ReconViaGen exhibits strong robustness to moderate cross-view inconsistency.

---

### Author Response · Authors · 2025-11-26
**General Response**

We thank all reviewers for their detailed reviews and suggestions!

We have updated the manuscript with the following revisions based on the reviewers' suggestions. All revisions in the updated manuscript are highlighted in red:

**Add more experiments**
- Add quantitative comparison of generated multi-view images on the Dora-bench dataset (reviewer Zbjq, Appendix A.6 line 916-955, Table 8).
- Add comparison experiment of single-view input on the Dora-bench dataset (reviewer Zbjq, Appendix A.8 line 1002-1012, Table 9 and Figure 8).
- Add more visual comparisons for (c) vs (d) in ablation study. (reviewer LmLx, Appendix A.9 line 1015-1019, Figure 9).
- Add ablation study on hyperparameters of RVC (reviewer eFoP and LmLx, Appendix A.10 line 1022-1052, Table 10, Table 11 and Table 12).
- Add the detailed latency of each component in ReconViaGen (reviewer eFoP, LmLx and 3uhU, Appendix A.11 line 1055-1063, Table 13).
- Add ablation study on decoding to different 3D representations in RVC (reviewer eFoP, Appendix A.13 line 1100-1114, Table 14).
- Add attention visualizations in GGC and cross-view token similarity in PVC (reviewer eFoP, Appendix A.14 line 1115-1126, Figure 10 and Figure 11).
- Add ablation study on the length of learnable tokens in GGC (reviewer eFoP, Appendix A.15 line 1115-1126, Table 15).
- Add evaluation on the DTU dataset (reviewer eFoP, Appendix A.16 line 1173-1180, Table 16 and Figure 12).
- Add ablation study on RVC only baseline with our full-version model (reviewer LmLx, Appendix A.17 line 1183-1187, Table 17).
- Add ablation study on alternative design for weighted fusion in SLat-Flow (reviewer LmLx, Appendix A.18 line 1231-1241, Table 18).
- Add sensitivity analysis of the quality to camera pose accuracy (reviewer 3uhU, Appendix A.20 line 1296-1302, Table 19).


**Add more discussion and detials**
- Add the training difference between SS-Flow and SLat-Flow (reviewer LmLx, line 357-359).
- Add the interpretability of GGC and PVC (reviewer eFoP, Appendix A.5, line 897-903).
- Add the theoretical explanation on how RVC interacts with rectified flow objectives (reviewer eFoP, Appendix A.12, line 1067-1080).
- Add clarification for omitting the visual metrics of VGGT in the Dora-Bench evaluation (reviewer LmLx, Appendix A.19, line 1273-1283).

**Change notation to avoid confusion**
- Change the mathematical notation of the two ConditionNets into two different symbols (reviewer LmLx, line 248-269).

Thanks again for all the effort and time, and we look forward to further discussions if there are any more questions.

---

> ### Author Response · Authors · 2025-12-01
> **Further Summary**
>
> Dear reviewers and AC,
>
> We sincerely thank all reviewers for their valuable feedback and encouraging comments regarding our **novel and effective** model (Reviewer Zbjq, 3uhU), **significant improvements** (Reviewer Zbjq, eFoP, LmLx), **strong motivation** (Reviewer eFoP, 3uhU), **comprehensive evaluation** (Reviewer eFoP, 3uhU), **innovative and well-designed framework** (Reviewer eFoP,  3uhU) and **clear writing** (Reviewer Zbjq, eFoP, LmLx, 3uhU).
>
> Before the discussion period was interrupted, two reviewers provided updated feedback:
>
> Reviewer LmLx upgraded the rating from 4 to 6, affirming that our efforts and detailed clarifications for the concerns and believing that our work will inspire the future researchers to think better in terms of connecting transformer based pipelines through clean modules, into achieving robust and useful applications (26 Nov 2025, CST 17:23).
>
> Reviewer 3uhU maintained the initial score at 6, noting that our reply and extended experiments addressed the concerns and tend to accept this work (26 Nov 2025, CST 17:35).
>
> After the dscusion was interuped, one reviewer provided updated feedback:
>
> Reviewer eFoP maintained the initial score at 6, noting that our detailed responses addressed the concerns and inclined to accept this work (28 Nov 2025, CST 11:33).
>
> For the last reviewer who did not provide later comments, we have already provided detailed rebuttals to address the concerns.
>
> We sincerely hope that our analysis and clarifications address all concerns.
>
> Thank you for your time and consideration.
>
> Best regards,
>
> Authors

---

### Meta-Review · Area_Chair_YJaR · 2025-12-30

**Summary:**

The paper presents an approach that combines the reconstruction prior in VGGT with a generative prior in TRELLIS, performing unconstrained, generative reconstruction while correctly respecting the input observations. The reviewers highlight the strong motivation, the novel and effective model and the good evaluation. I agree, I think this is a very relevant problem solved and like the approach. The reviewers gave the paper positively-leaning borderline scores in the first round (4, 6, 6, 6), while stating several concerns that are mostly about potential experiments that could be performed. The authors delivered strongly in the discussion phase, providing a wide variety of new results that address almost all stated concerns. As a result, the negatively-leaning reviewer promised to increase the score to 6. All in all, I decided to follow the reviewers recommendation and recommend to accept the paper. Congratulation to the authors to this great work!

**Reviewer Concerns:**

*1) No single view input.* The authors provide an additional experiment with single view input, showing that the method works reasonably well.

*2) Robustness to inconsistent inputs.* The authors provide a robustness study for this case.

*3) Insufficient Hyperparameter analysis.* The authors provide an additional study analyzing effect of hyperparameters.

*4) Potential train-inference gap.* Addressed by additional discussion and experiments.

*5) Pose refinement pipeline is engineering-heavy.* Alternatives where discussed and deemed worse by the authors. I find this sufficient, as I made similar observations.

*6) RVC only baseline missing.* Provided by the authors, the full method outperforms the baseline.

*7) Not enough qualitative examples.* Provided by the authors.

*8) No detailed analysis of inference speeds.* Provided by the authors.

*9) Unclear robustness of pose estimation.* Investigated by the authors and results provided.

In addition to the above, the authors provided several additional experiments, which where not necessarily addressing weaknesses but where asked for by the reviewers.

**Reviewer Scores:**

The only negatively reviewer (Reviewer LmLx, score 4) already confirmed to raise the score (but was not able to). Assuming the others remain, this would lead to scores of (6, 6, 6, 6).

---

### Decision · Program_Chairs · 2026-01-26

Accept (Poster)